# HIPP neurons in the dentate gyrus mediate the cholinergic modulation of background context memory salience

Syed Ahsan Raza[1], Anne Albrecht[1], Gürsel Çalışkan[1], Bettina Müller[1], Yunus Emre Demiray[1], Susann Ludewig[1], Susanne Meis[2], Nicolai Faber[3], Roland Hartig[4], Burkhart Schraven[5,6], Volkmar Lessmann[2,7], Herbert Schwegler[3,7] & Oliver Stork[1,7]

Cholinergic neuromodulation in the hippocampus controls the salience of background context memory acquired in the presence of elemental stimuli predicting an aversive reinforcement. With pharmacogenetic inhibition we here demonstrate that hilar perforant path-associated (HIPP) cells of the dentate gyrus mediate the devaluation of background context memory during Pavlovian fear conditioning. The salience adjustment is sensitive to reduction of hilar neuropeptide Y (NPY) expression via dominant negative CREB expression in HIPP cells and to acute blockage of NPY-Y1 receptors in the dentate gyrus during conditioning. We show that NPY transmission and HIPP cell activity contribute to inhibitory effects of acetylcholine in the dentate gyrus and that M1 muscarinic receptors mediate the cholinergic activation of HIPP cells as well as their control of background context salience. Our data provide evidence for a peptidergic local circuit in the dentate gyrus that mediates the cholinergic encoding of background context salience during fear memory acquisition.

[1] Department of Genetics and Molecular Neurobiology, Institute of Biology, Otto-von-Guericke University, 39120 Magdeburg, Germany. [2] Institute of Physiology, Medical Faculty, Otto-von-Guericke University, 39120 Magdeburg, Germany. [3] Institute of Anatomy, Medical Faculty, Otto-von-Guericke University, 39120 Magdeburg, Germany. [4] Core Facility Multidimensional Microscopy and Cellular Diagnostics at the Medical Faculty, Otto-von-Guericke University, 39120 Magdeburg, Germany. [5] Institute of Molecular and Clinical Immunology, Medical Faculty, Otto-von-Guericke University, 39120 Magdeburg, Germany. [6] Department of Immune Control, Helmholtz-Centre for Infection Research, 38124 Braunschweig, Germany. [7] Center for Behavioral Brain Sciences, 39106 Magdeburg, Germany. Syed Ahsan Raza and Anne Albrecht contributed equally to this work. Correspondence and requests for materials should be addressed to O.S. (email: oliver.stork@ovgu.de)

Contextual information provides a unique temporal and spatial signature for a given experience, specifying the "where" and "when" of episodic memories. In simple conditioning paradigms, when a discrete stimulus (a conditional stimulus, CS) is associated with a coinciding unconditional stimulus (US), a stable set of features that form such a context can be conditioned together with the elemental CS[1]. If the elemental CS lacks significant predictive value for the US, the contextual information is processed as "foreground context" and can induce full memory retrieval upon subsequent exposure. By contrast, when an elemental CS is systematically paired with the US, incidental learning of the conditioning context occurs. As a result, the contextual information is processed as "background context" and provides an episodic representation for the memorized experience that determines its strength and specificity[2]. When presented without the elemental CS, however, such background context memory evokes only a partial response[1–3] (see Fig. 1a, b). Previous studies have established a circuit from the medial septum to the hippocampus in controlling such background context memory[4, 5]; however, the involved intrahippocampal circuits and their relation to these cholinergic septal afferences have not been resolved so far.

Converging evidence now suggests that a local inhibitory circuit in the dorsal dentate gyrus (DG), formed between granule cells and hilar perforant path-associated (HIPP) cells, may be involved in adjusting the level of background context fear.

First, it is firmly established that the hippocampus is critical for the integration of multimodal information into higher-order representations of context, as well as for context discrimination[6]. This hippocampal function is specified by the presence or absence of an elemental CS[7, 8]. Phillips and LeDoux (1994) showed that lesions of the dorsal hippocampus interfere with background context memory, whereas lesions of the ventral hippocampus interfere with foreground context memory. Correspondingly, molecular activity markers in the intact hippocampus are differentially increased in the dorsal and ventral regions upon background and foreground context conditioning[9].

Second, the DG is indispensable for context memory formation and critically involved in the discrimination of similar contexts[10, 11]. Evidence suggests that the DG granule cells can be incorporated into a context memory engram[12] and that the stimulation of certain preformed DG granule cell ensembles is both necessary and sufficient to activate a corresponding non-reinforced contextual memory[13, 14]. The size of activated DG granule cells ensembles correlates with context memory strength[15] and their output modulates the encoding of strength and specificity of context fear memories in area CA3[10, 14].

Third, the activity of granule cells during memory encoding is orchestrated by HIPP cells. These are GABAergic local circuit neurons expressing somatostatin (SST) and neuropeptide Y (NPY) as co-transmitters. With their somata located in the hilus proper, they receive inputs from granule cells via mossy fiber collaterals and directly from the perforant path, and provide inhibition to the outer molecular layer of the DG[16–18]. Thus they contribute to negative feedback and feed-forward circuits that modulate the firing rate of granule cells, prevent excessive DG discharge and modulate the formation of granule cell ensembles during conditioning[13].

In the current study we set out to examine the putative role of HIPP cells in the modulation of background context salience and to identify critical molecular factors that define this function of HIPP cell circuitry in the dentate gyrus.

## Results

### HIPP cell activity controls background context fear.
First, we confirmed in C57Bl/6 mice that our training paradigms generate

with identical sensory stimuli significantly reduced levels of contextual memory following background context conditioning, compared to foreground context conditioning. Further, we ensured that the latter induced full contextual salience as defined by the level achieved with pure context conditioning without an elemental CS. Our training protocols thus produced significantly different levels of freezing behavior during contextual (One-way ANOVA: F (2.21) = 10.96, $P = 0.001$; $n = 8$ each) and auditory cued (One-way ANOVA: F (2.21) = 16.77, $P < 0.001$; $n = 8$ each) memory retrieval (Fig. 1a–b).

Then, to examine the putative role of HIPP cells in the encoding of background context memory we targeted them with adeno-associated virus hSyn-DIO-hM4Di-mCherry (Fig. 1c), which allows for transient silencing upon systemic administration of clozapine-N-oxide (CNO)[19]. The adeno-associated virus hSyn-DIO-mCherry, which conditionally expresses only the fluorescent marker, served as a control. Upon injection of viruses into the hilus of SST–Cre[ERT2] driver mice and activation of CRE recombinase with tamoxifen, HIPP cells were effectively and specifically targeted for the pharmacogenetic manipulation (Fig. 1d, f, g and Supplementary Fig. 1d). Their inactivation during memory acquisition led to a significant increase of contextual memory following training with paired CS-US presentations (context in background) (Fig. 1e). By contrast, no effect was observed on foreground context memory, i.e. when conditioning was done with unpaired presentations of CS and US (Fig. 1e) or without any CS presentation (Supplementary Fig. 2a, b). Thus, HIPP cell inactivation abolished the differences in contextual freezing between training protocols.

By contrast, the pharmacogenetic inhibition had no effect on freezing levels during auditory cued memory retrieval, which remained high in the (paired) background context group and low in the (unpaired) foreground context group (Fig. 1e), as well as in pure context memory group (Supplementary Fig. 2a, b). Furthermore, pharmacogenetic inhibition of HIPP cells during retrieval was without effect on background context memory or auditory cued memory (Supplementary Fig. 2c, d). The behavior during pre- and immediate post-training intervals and upon exposure to a neutral context was also not affected by the viral intervention (Supplementary Fig. 1b, c). In summary, we observed a selective enhancement of background context memory by HIPP cell inactivation during training.

HIPP cell inactivation is likely to induce a disinhibition of DG granule cells, as well as parvalbuminergic basket cells[17] and their pharmacogenetic silencing has previously been shown to increase cFos induction by contextual fear conditioning in DG granule cells[15]. Accordingly, we found that silencing HIPP cells with hM4Di during background training also increased the population of cFos-expressing cells in the DG 1 h after background context conditioning (Fig. 1h, i). Parvalbumin interneurons provide an important class of local circuit neurons in the dentate gyrus that exerts powerful control over granule cell activity[20] and is modulated in firing precision through HIPP neurons[17]. To mimic a potential disinhibition of these cells through HIPP cell silencing, we activated DG basket cells via hSyn-DIO-hM3Dq-mCherry in PV-Cre transgenic mice during training (Supplementary Fig. 3a–d). However, this intervention did not increase, but reduced the background context memory without affecting auditory cued memory or behavior during training or neutral context exposure.

### CREB controls NPY expression and background context fear.
CREB-expressing viral vectors have been repeatedly used to identify and manipulate neurons forming fear memory engrams[21] and contextual fear memory has been blocked through the viral

expression of a dominant negative form of CREB in the dorsal hippocampus[22]. Therefore, to begin addressing the intracellular signaling events in HIPP cells, we generated a lentiviral vector for the conditional expression of dominant negative CREB (CREB[S133A]; Fig. 2a, e, f and Supplementary Fig. 4c). In line with

the effect of hM4Di, the expression of CREB[S133A] in the hilus of SST-Cre[ERT2] mice led to an increase of background context memory as compared to mutants injected with control viruses (Fig. 2b, c). The responses to the CS and neutral context, pre- and immediate post-training freezing and anxiety-like behavior were

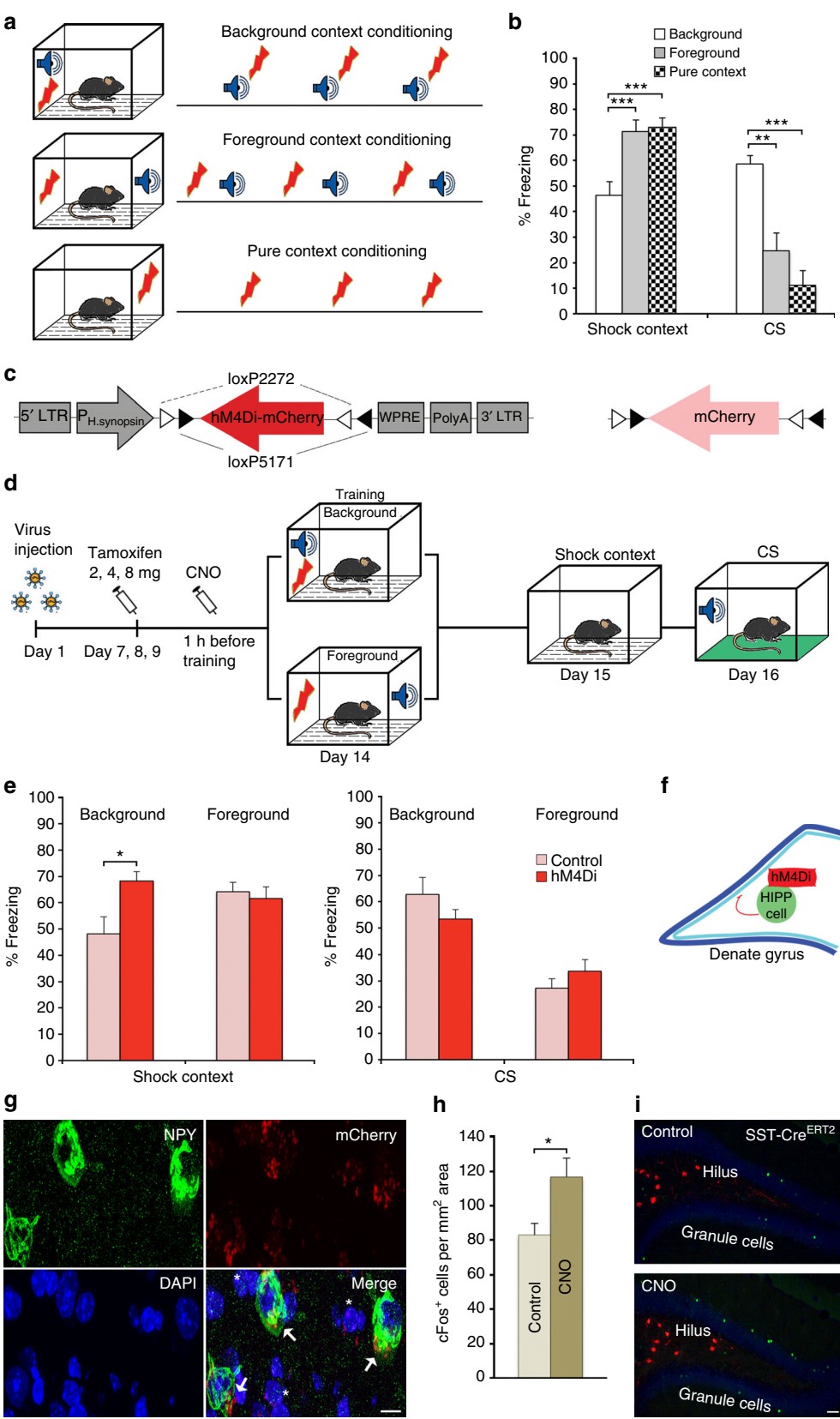

not different between these groups (Supplementary Fig. 4a, b). Thus, although CREB$^{S133A}$ expression can be expected to evoke a continuous perturbation of cellular function, its behavioral effects closely resembled that observed after HIPP cell inhibition during memory acquisition.

For the effectiveness and specificity of this viral manipulation, we confirmed mRNA expression of the virally transduced CREB and measured expression levels of the HIPP−cell markers and putative CREB targets, SST, NPY and glutamic acid decarboxylase (GAD)67. Expression of CREB$^{S133A}$ in HIPP cells reduced the expression of NPY mRNA in the hilus, whereas overexpression of native CREB in these cells increased NPY mRNA levels (One-way ANOVA: F $(2.16) = 30.221$; $P = 0.001$; $n = 6$ each; Fig. 2d and Supplementary Fig. 4d). The viral manipulation also affected SST (One-way ANOVA: F $(2.15) = 4.507$; $P = 0.03$; $n = 6$ each) and GAD67 mRNA levels (One-way ANOVA: F $(2.16) = 18.595$; $P = 0.001$; $n = 6$ each), but while both viral vectors increased SST expression only native CREB enhanced GAD67 (Fig. 2d).

**Background context conditioning activates CREB in HIPP cells.** Increased CREB phosphorylation as an indicator of cellular activity has been described for all sub-regions of the hippocampus after fear conditioning[23]. Given the apparent relation between CREB activity and NPY expression in HIPP cells, we tested if CREB activation might occur in NPY$^+$ cells after background context conditioning. Approximately two-third of the hilar SST$^+$ cells in our experimental animals co-expressed NPY, which was on the other hand virtually absent from SST$^-$ cells in the hilus (Supplementary Fig. 1f). Congruent double labeling of the transgenic GFP with virally expressed mCherry was confirmed in SST-Cre$^{ERT2}$/NPY-GFP double mutant mice (Supplementary Fig. 1g). Therefore, NPY-GFP mutants were further on used for the identification of NPY$^+$ HIPP cells.

Strikingly, we observed an increase in pCREB$^{S133}$/GFP double labeling in the hilus of NPY-GFP mice 1 h after background context conditioning (One-way ANOVA: $0.222 \pm 0.0162$ cells per 1000 μm$^2$, $P < 0.02$; $n = 6$ each), but not after foreground context conditioning ($0.185 \pm 0.013$ cells per 1000 μm$^2$), compared to control levels ($0.150 \pm 0.016$ cells per 1000 μm$^2$). Training significantly influenced the number of double-labeled cells (pCREB$^+$/GFP$^+$) at this time point in the hilus (One-way ANOVA for the effect of treatment: F $(2.17) = 9.55$; $P = 0.002$; $n = 6$ each; Fig. 2g, h). By contrast, the conditioning protocols did not have an observable effect on the number of pCREB$^+$/GFP$^+$ labeled cells in the *stratum oriens* of the CA1, which was evaluated on the same sections to control for regional specificity (One-way ANOVA: F $(2.17) = 0.11$; *n.s.*; $n = 6$; Supplementary Fig. 4e).

**NPY in the DG controls background context fear.** Based on these observations we next examined the putative role of NPY in the control of background context memory. The major post-synaptic receptor for NPY in the DG is the Y1 receptor, which mediates the NPY-induced inhibition of both granule cells and hilar interneurons[24]. Infusion of the Y1 receptor antagonist BIBP3226 bilaterally into the dorsal DG before fear conditioning training increased the background context memory after CS−US pairing without affecting fear memory to the CS or behavior during training. By contrast, BIBP3226 infusion did not affect foreground context memory, induced by unpaired training with the same stimuli (Fig. 3a, d and Supplementary Fig. 5a, b). BIBP3226 injection before retrieval was without effect on background context memory, indicating a specific role of NPY in the acquisition of fear memory (Supplementary Fig. 5c, d). This function appears to be independent of the anxiolytic properties of NPY, since we found no change in background conditioning upon targeting Y1 receptors with intracerebroventricular injections of a higher, anxiogenic dose of BIBP3226 (Supplementary Fig. 5e, f).

**M1 receptors control the activity of hilar NPY$^+$ cells.** To identify the pathways that activate HIPP cells and NPYergic transmission during background context conditioning, we examined the mRNA expression of various neuromodulatory receptors in hilar NPY$^+$ interneurons. To this end, we used behaviorally naive NPY-GFP mice and isolated the GFP$^+$ cells from the hilus and from the hippocampal areas CA1 and CA3, as well as the GFP$^-$ rest of the hilar tissue, using laser capture microdissection (Supplementary Fig. 6a). Quantitative PCR analysis confirmed the enrichment of NPY mRNA in the hilar and cornu ammonis GFP$^+$ neurons compared with GFP$^-$ hilus samples (One-way ANOVA for cell population effect: F $(2.18) = 12.362$; $P < 0.001$; $n = 6–8$; Fig. 4a). Cell type specific expression levels were further observed for SST mRNA (One-way ANOVA: F $(2.21) = 6.46$; $P = 0.007$; $n = 6–8$) and GAD67 mRNA (One-way ANOVA: F $(2.18) = 25.094$; $P < 0.001$; $n = 6–8$), both of which were found at increased levels in hilar GFP$^+$ cells (Supplementary Fig. 6b), confirming the expected neurochemical HIPP cell signature.

Further analysis revealed a cell type specific expression of the M1 subtype of muscarinic acetylcholine receptors (Chrm1) (One-way ANOVA: F $(2.19) = 3.677$; $P = 0.04$; $n = 6–8$) that was increased in GFP$^+$ neurons compared to in GFP$^-$ neurons of the hilus (Fig. 4a; see also Supplementary Table 2). By contrast, Chrm2 was not detected in hilar GFP$^+$ neurons (One-way ANOVA: F $(2.19) = 9.940$; $P = 0.001$; $n = 6–8$; Fig. 4a), which is in agreement with the reported specific localization of this receptor

**Fig. 1** Pharmacogenetic inactivation of HIPP cells increases background context conditioning. **a** Schematic of the context conditioning paradigms. **b** Contextual and cued fear memory of C57Bl/6 mice trained according to those protocols. The contextual fear response is significantly lower in the background context group ($n = 8$) than in both foreground context ($n = 8$) and pure context ($n = 8$) groups in agreement with previous reports [4, 5, 9]. At the same time, freezing to the CS is higher in the background context group, as expected. **c** Conditional hM4Di-mCherry and mCherry-control vectors. **d** Schematic of viral transduction and behavioral testing paradigms, depicting the virus injection, the activation of CRE recombination with increasing doses of tamoxifen and the activation of hM4Di with clozapine-N-oxide (CNO). Background and foreground context conditioning were applied to different groups and contextual and auditory memory was tested in both 24 h and 48 h later, respectively. **e** (*Left*) Memory to the background context is increased when HIPP cells are silenced during background context conditioning in SST-Cre$^{ERT2}$ mice ($n = 8$; controls $n = 10$), whereas no effect is observed upon HIPP cells silencing during foreground conditioning ($n = 8$; controls $n = 7$). (*Right*) As expected the CS response is high in the background context group compared to the foreground context group. Viral intervention has no effect on cued memory, as the response to auditory stimuli remains unaltered in both training paradigms. **f** A schematic of the proposed circuitry. **g** Representative microscopic images show the adenoviral expression of mCherry-tagged hM4Di receptors in NPY$^+$ cells (*arrows*) and NPY$^-$ cells of the hilus (*asterisks*). For an overview of the viral expression please see Supplementary Fig. 1d. Scale bar, 10 μm. **h** The number of cFos$^+$ granule cells is increased after background context conditioning if hM4Di targeted HIPP cells are silenced with CNO ($n = 6$ each). **i** Representative images depicting cFos labeling and mCherry-tagged hM4Di in the DG following background context conditioning. Scale bar, 50 μm. Data are means + s.e.m. Statistical analysis was done with Fisher's LSD following one-way ANOVA in **a** and with Student's unpaired $t$-test in **e**, **h**. *$P < 0.05$; **$P < 0.01$; ***$P < 0.001$

type on the dendrites of parvalbuminergic basket cells in this area[25]. We confirmed the previously reported rich cholinergic fiber innervation into hilar NPY[+] cells[26] (Fig. 4b) and demonstrated the expression of M1 receptors on the cell body of GFP[+] interneurons in NPY–GFP mice (Fig. 4c and Supplementary Fig. 6c). Functionally, M1 receptors mediate a muscarinic excitation of these interneurons, as shown by patch clamp recordings and the blockage of M1 receptors via the selective antagonist, pirenzepine (Fig. 4d).

Using acute slice preparations of the dorsal hippocampus, we tested whether NPY mediates M1 receptor function in the DG. We found that neither Y1 receptors blockade nor HIPP cells silencing had an effect on baseline excitability, as seen by unaltered population spike (PS) area (Supplementary Fig. 7a).

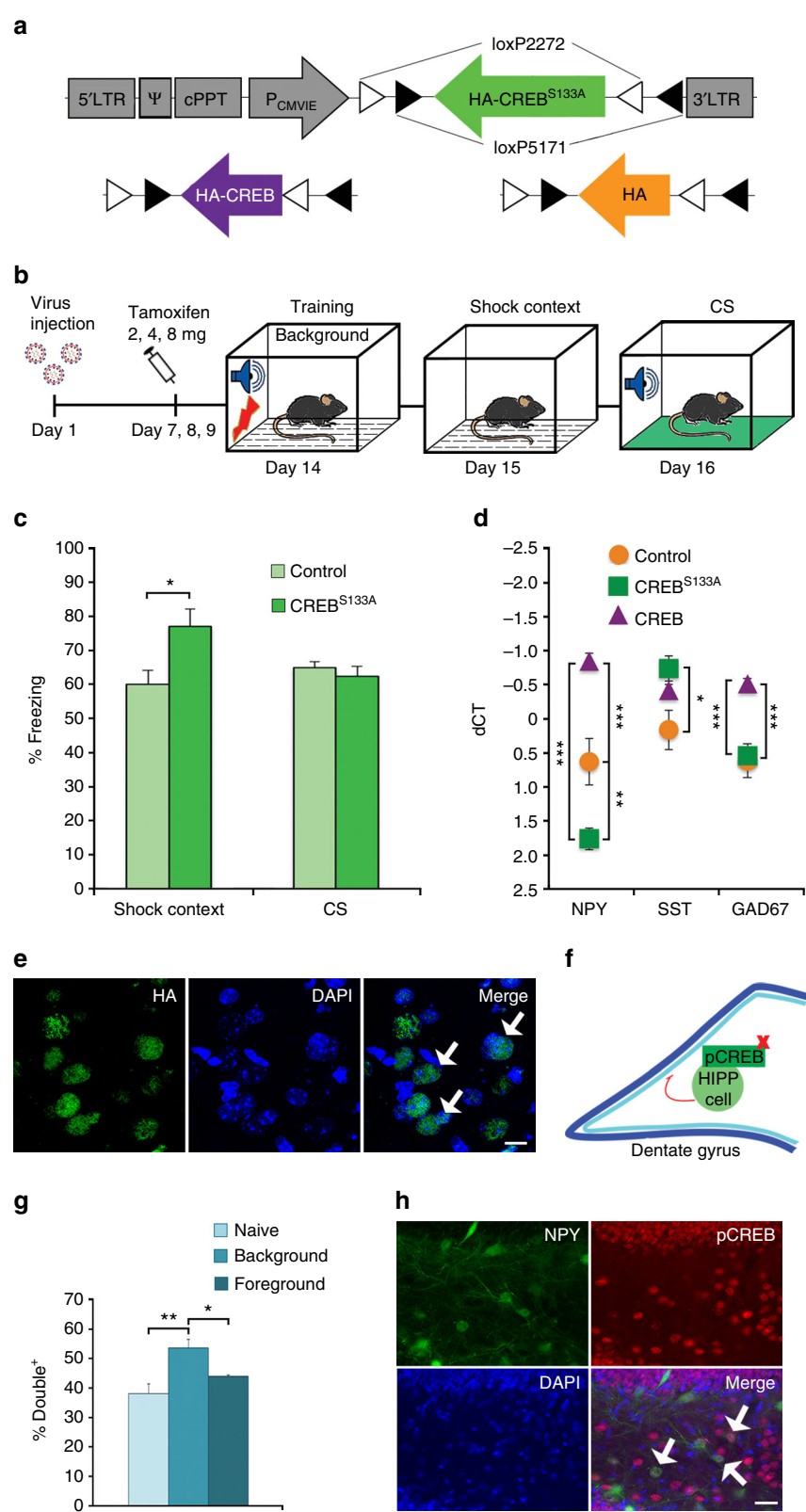

However, under conditions of increased muscarinic activation by slice perfusion with 1 μM oxotremorine M (oxo), which itself led to a significant reduction in PS area in control slices (Fig. 4e), a strong interaction was observed. Blockage of Y1 receptors with 1 μM BIBP3226 prior to oxo perfusion strongly counteracted the oxo-mediated inhibition resulting in increased PS area compared to control slices (Fig. 4e). This BIBP3226 effect was mimicked by pharmacogenetic inhibition of HIPP cells in SST-Cre$^{ERT2}$ mice expressing hM4Di viruses (Fig. 4e). Similar effects were observed in combination with 10 μM carbachol instead of oxo (Unpaired $t$-test: BIBP3226: t (16) = −2.353; $P < 0.05$; $n = 8$ slices each; CNO: t (14) = −3.296; $P < 0.01$; $n = 10$ slices each).

Next, we tested with an occlusion experiment for potential NPY-independent components in the effect of HIPP cell silencing. However, when comparing the oxo-induced depression effect in slices of hM4Di transduced SST-Cre$^{ERT2}$ driver mice treated only with BIBP3226 and those treated with both BIBP3226 and CNO, no difference could be found (Supplementary Fig. 7b).

We hypothesized that attenuation of CREB activity in HIPP cells might disturb this circuit function and therefore examined PS in the slices obtained from SST-CRE$^{ERT2}$ driver mice expressing CREB$^{S133A}$. To determine potential changes in HIPP cell sensitivity, two different concentrations of oxo (0.1 μM and 1 μM) were applied in the presence or absence of BIBP3226. Upon application of a low concentration of oxo an effect of CREB$^{S133A}$ expression was evident. Comparison between groups confirmed significant main effects of CREB$^{S133A}$ expression (Two-way ANOVA: F (1.25) = 5.013; $P = 0.034$; $n = 5$–9) and BIBP3226 treatment (Two-way ANOVA: F (1.25) = 13.543; $P = 0.001$; $n = 5$–9), as both decreased the oxo-mediated inhibition. However, the effect of CREB$^{S133A}$ was less pronounced than that of BIBP3226 and no interaction between these factors was found (Two-way ANOVA: F (1.25) = 0.291; n.s; $n = 5$–9). Using the higher concentration of oxo, a BIBP3226 effect was again observed (Two-way ANOVA: F (1.25) = 5.711; $P = 0.025$; $n = 5$–9), but no effect of CREB$^{S133A}$ (Two-way ANOVA: F (1.25) = 0.016, n.s.; $n = 5$–9) and no interaction between these factors became evident (Two-way ANOVA: F (1.25) = 0.006, n.s.; $n = 5$–9; Supplementary Fig. 8).

**HIPP cell M1 receptors control background context salience.** Based on these findings we investigated the behavioral consequence of M1 receptor manipulations in the DG. Pharmacological blockage of DG muscarinic receptors with scopolamine suppressed the context fear response (One-way ANOVA: F (2.24) = 7.861, $P = 0.002$; $n = 6$ for 2.5 μg μl$^{-1}$, $n = 11$ for 5 μg μl$^{-1}$, $n = 10$ for vehicle; Supplementary Fig. 9a), in line with findings in other tasks[27], but did not significantly affect cued memory (One-way ANOVA: F (2.24) = 2.50, $P = 0.1$). Scopolamine injection further suppressed immediate post training freezing (One-way ANOVA: F (2.24) = 49.46, $P < 0.001$; $n = 6$ for

2.5 μg μl$^{-1}$, $n = 11$ for 5 μg μl$^{-1}$, $n = 10$ for vehicle; Supplementary Fig. 9a), and dose dependently altered activity and anxiety-like behavior in an elevated plus maze (Kruskal–Wallis test: total arm entries: H (2) = 13.53, $P = 0.001$; One-way ANOVA: % open arm entries: F (2.23) = 13.14, $P < 0.001$; $n = 7$ for 2.5 μg μl$^{-1}$, $n = 10$ for 5 μg μl$^{-1}$, $n = 9$ for vehicle; Supplementary Fig. 9b).

As scopolamine injection likely affected also muscarinic receptors that are expressed on the DG granule cells and other hilar interneurons, we decided to selectively perturb the M1-receptor mediated control of HIPP cell activity. To this end we developed lentiviral vectors for the conditional knockdown of Chrm1 and injected them into the dorsal hilus of SST-Cre$^{ERT2}$ driver mice (Fig. 5a–c). Indeed, the co-expressed DsRed marker was found in NPY$^+$ and some NPY$^−$ cells of the hilus, but not in the granule cell layer (Fig. 5d). In line with the effects of the pharmacogenetic inhibition of HIPP cells and of the Y1 receptors blockage, a significant increase in background context memory was observed in these animals (Fig. 5e). No change was evident in the response to CS presentation, clearly distinguishing the context specificity of HIPP-cell directed Chrm1 knock down. Moreover, no effect of this manipulation was found on animal behavior during habituation or testing in the neutral context, but an enhancement of immediate post training freezing was evident that resembles the observed change in response to the background context (Supplementary Fig. 9d).

**Endogenous ACh release recruits NPY transmission in the DG.** Therefore we finally asked how the pharmacologically and pharmacogenetically identified HIPP cell circuit might contribute to the effects of endogenous ACh release in the DG. To this end, we expressed hM3Dq in septo-hippocampal projection neurons of Chat-Cre driver mice that terminate in the dorsal DG (Fig. 6a, b) and selectively activated their cholinergic terminals in our slice preparations through bath application of 10 μM CNO. This activation resulted in a moderate increase of the DG PS area, which was augmented by blockage of Y1 receptors with BIBP3226 (1 μM) (One-way repeated measures ANOVA: F (2.14) = 28.701, $P < 0.001$; $n = 8$ slices; Fig. 6c, d). Importantly, no effect of BIB3226 on the PS responses was apparent in the absence of CNO (Fig. 6e and Supplementary Fig. 7a). When comparing the isolated BIBP3226 effect between experiments a significant increase was evident under cholinergic activation with CNO (Fig. 6f).

## Discussion

Fundamental mechanisms of memory storage have been clarified over the last years and the principles of neuronal ensemble formation during learning are beginning to unfold. We here describe a local NPYergic circuit formed by HIPP cells in the DG that mediates the cholinergic modulation of background context salience during Pavlovian fear conditioning (Fig. 7). By evaluating

**Fig. 2** CREB activation in HIPP cells during background context conditioning. **a** Lentiviral vector constructs for the conditional expression of control (HA), native CREB and dominant negative CREB$^{S133A}$. **b** Schematic of the viral transduction and behavioral testing paradigm. **c** CREB$^{S133A}$ ($n = 8$) increases fear memory to the background context, while the fear response to the CS remains unaltered (controls $n = 10$). **d** The overexpression of native CREB in SST-Cre$^{ERT2}$ mice increases, whereas dominant negative CREB$^{S133A}$ reduces NPY mRNA levels in the hilus ($n = 6$ each). GAD67 mRNA is also increased by CREB overexpression. SST mRNA by contrast is increased in the CREB$^{S133A}$ group with a non-significant trend for native CREB, indicative of a phosphorylation-independent constitutive expression regulation. Note the inverted scaling of the ordinate, the lower values indicate higher expression. **e** A representative microscopic image shows the lentiviral expression in the hilus. *Green*, HA; *blue*, DAPI. Scale bar, 10 μm. For an overview, please see Supplementary Fig. 4c. **f** A schematic of the proposed circuit and the site of CREB manipulation. **g** The proportion of pCREB$^{S133}$/GFP double positive cells in the hilus of NPY-GFP transgenic mice is selectively increased 1 h after background context conditioning ($n = 6$ each). **h** Representative microscopic images show phosphorylated CREB expression in the hilus and granule cell layer of the DG. *Arrows*, pCREB$^{S133}$/GFP double-labeled cells. Scale bar, 25 μm. Data are means + s.e.m. Statistical analysis was done with Student's unpaired $t$-test in **c**, and by Fisher's LSD following one-way ANOVA in **d** and **g**. *$P < 0.05$; **$P < 0.01$; ***$P < 0.001$

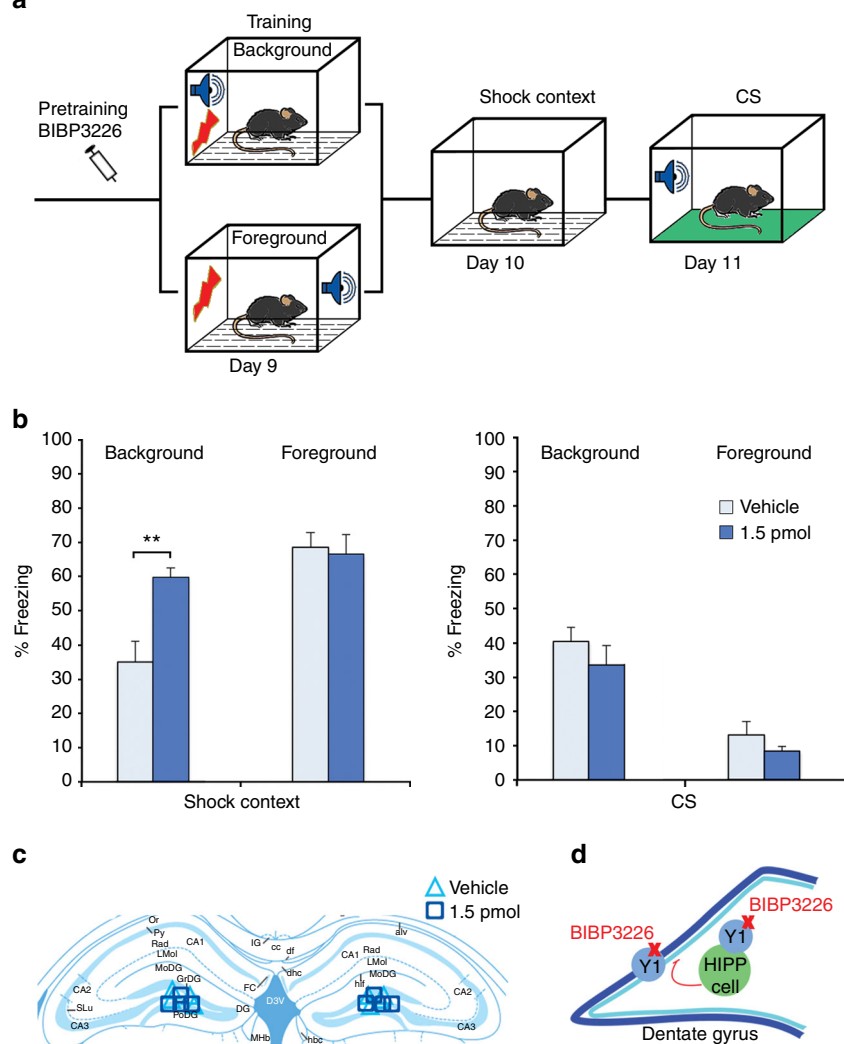

**Fig. 3** NPY in the dorsal DG controls background context fear memory. **a** A schematic of the behavioral paradigms. **b** Blocking Y1 receptors with BIBP3226 (1.5 pmol per hemisphere; $n = 9$) during training increases background context conditioning compared to vehicle ($n = 11$). BIBP3226 treatment does not induce any change, when used in foreground context conditioning (vehicle $n = 7$; 1.5 pmol BIBP3226 $n = 8$). No effect is observed on auditory fear memory, as the background context group maintains unchanged cue responding after injection, while the foreground context group continues to show only low levels of freezing. **c** Localization of guide cannulas in the dorsal DG. **d** A schematic of the proposed local circuit and site of drug activity. Data are means + s.e.m. Statistical analysis was done with Student's unpaired t-test. **$P < 0.01$

episodic aspects of aversive experience, this circuit may help to specify the strength and specificity of fear memories[2].

The activity of the dentate gyrus is tightly controlled by local inhibitory circuits[28, 29], which are highly adaptive and sensitive to stress[30]. One prominent group of interneurons that mediate this inhibition are the SST expressing HIPP cells, which mediate feedback and feed-forward inhibition in the DG and control the pattern of granule cell activation[15, 17]. HIPP cells are known to inhibit both granule cells and parvalbuminergic basket cells in the dentate gyrus; furthermore, they form a network through homosynaptic connections. HIPP cell activation reduces the excitability of granule cells and basket cells but also increases the firing precision of the later[17]. Our data now suggest that HIPP cells attenuate the activation of granule cells during background context encoding and that this inhibition reduces context salience.

HIPP cells are involved in a feedback loop with excitatory mossy cells, and computational modeling suggests that HIPP cells and mossy cells control the capacity of the DG for pattern-separation in opposite ways[18]. It has been suggested that this pattern-separation function of the DG critically contributes to the context specificity of memories[6] and the formation of conjunctive contextual representations[31]. Along this line, in order to discriminate full and partial context configurations during subsequent memory retrieval, the DG during fear memory acquisition must form a representation of the combined contextual and elemental stimulus configurations predicting the aversive reinforcement. The predictive value of a training situation loads onto both context and cue during fear conditioning; training-induced differences in context salience therefore are reflected in opposing changes in cue responding (see Fig. 1a, b, paired vs. unpaired).

By contrast, the manipulations of the context representation in HIPP cell circuitry in our experiments generally did not affect cued fear memory expression. This is in agreement with cholinergic control of background context conditioning: the pre-training inactivation of the medial septum by injection of lidocaine also enhances background context fear without having a significant effect on auditory cued memory[3]. Therefore it is likely

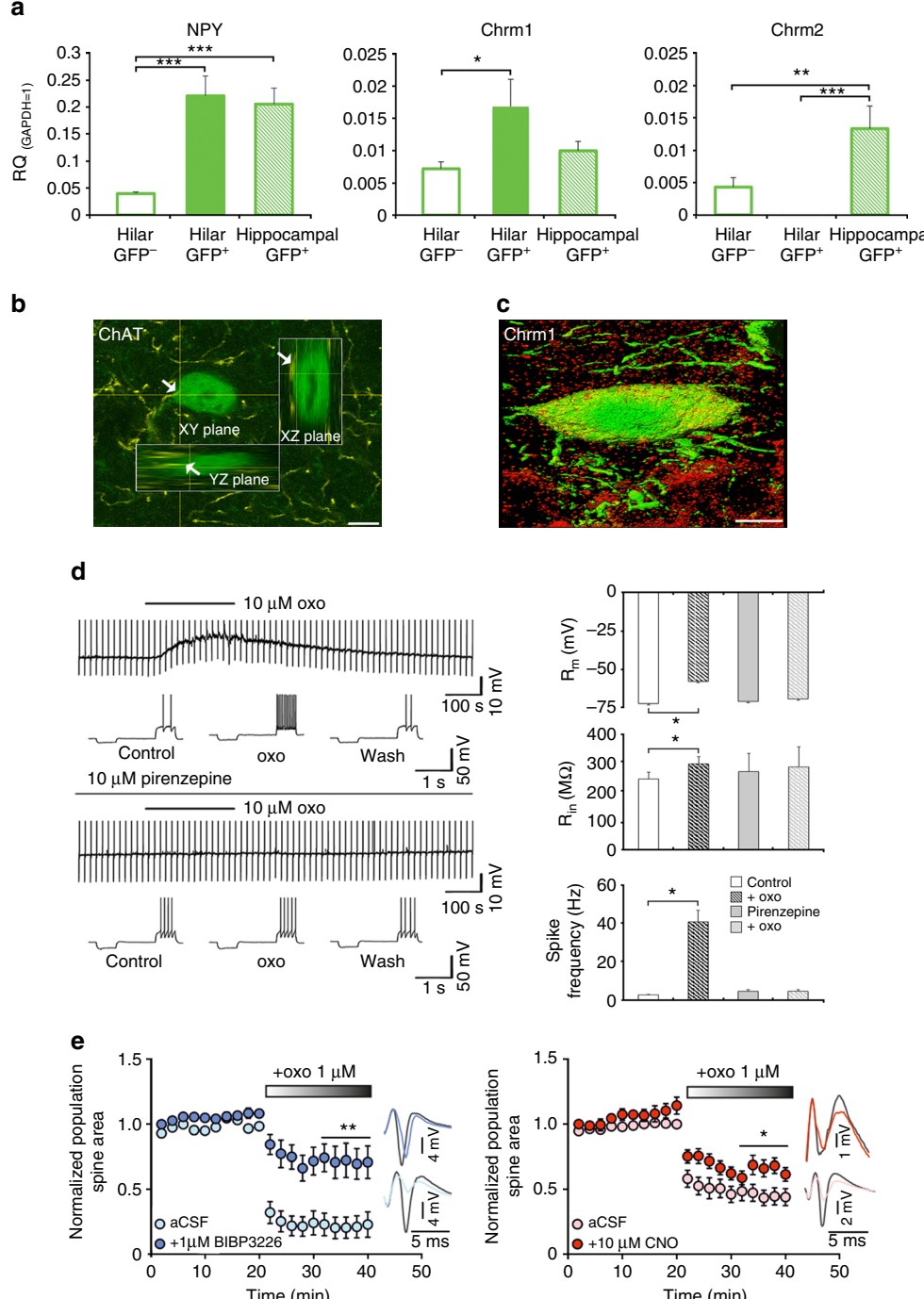

**Fig. 4** Chrm1 expression and function in HIPP cells. **a** Hilar GFP+ cells of NPY-GFP mice (*n* = 6–8) are enriched with mRNA for NPY and M1 muscarinic receptors (Chrm1) but not M2 receptor (Chrm2). Values are expressed relative to the housekeeping gene GAPDH. **b** A 3D-reconstruction of a GFP+ neuron targeted by cholinergic fibers (*arrows*) at the soma. Scale bar, 10 μm. **c** A GFP+ neuron (*green*) in the dorsal hilus of an NPY-GFP mouse with somatic immunolabeling of M1 receptors (*red*). Scale bar, 5 μm. **d** (*Left*) Representative current-clamp recording from GFP+ interneurons. Application of 10 μM oxotremorine M (oxo) induces a pirenzepine-sensitive transient depolarization paralleled by a significant increase in input membrane resistance and an increase in spike activity. (*Right*) Quantification of spike activity. Oxo induces a transient membrane depolarization from resting membrane potential in 7 out of 7 recorded neurons, paralleled by a significant increase in input membrane resistance and mean spike frequency elicited by positive current injections. These effects are abolished in the presence of the M1 receptor antagonist pirenzepine (10 μM; *n* = 5). **e** In field recordings, either Y1 receptor blockage (*left*) (*n* = 6 slices with BIBP3226 and *n* = 6 slices without BIBP3226) or pharmacogenetic HIPP cell inactivation (*right*) (*n* = 8 slices with BIBP3226 and *n* = 10 slices without BIBP3226) counteracts the depression of population spikes induced by oxo bath application. Example traces to the right of the graph illustrate the difference in response to perforant path stimulation. Values are means ± s.e.m. Statistical analysis was done with Fisher's LSD following one-way ANOVA in **a**, Wilcoxon signed rank test in **d** and Student's unpaired *t*-test in **e**. *$P < 0.05$; **$P < 0.01$; ***$P < 0.001$

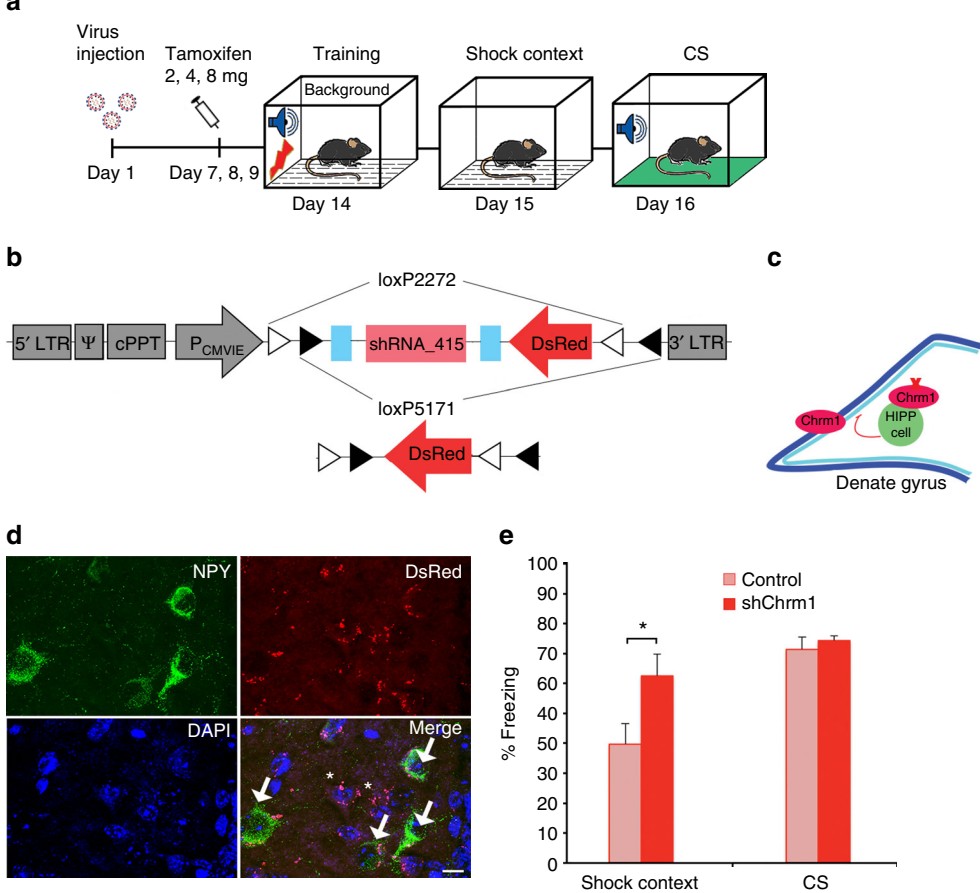

**Fig. 5** Chrm1 knock down in HIPP cells increases background context memory. **a** A schematic of the behavioral paradigm. **b** Lentiviral vector constructs for the conditional knockdown of Chrm1 and control vector. **c** Schematic of the proposed circuit. **d** Representative microscopic images show lentiviral expression of the DsRed marker in NPY[+] cells (*arrows*) and NPY[−] cells of the hilus (*asterisks*). Scale bar, 10 μm. For an overview refer to Supplementary Fig. 9f. **e** The knockdown of Chrm1 in SST-Cre[ERT2] mice ($n = 8$) significantly increases the memory to background context (controls $n = 7$), while the conditioned response to the CS remains unaffected. Data are means + s.e.m. Statistical analysis was done with Student's unpaired *t*-test. *$P < 0.05$

that the adjustment of background context salience via cholinergic modulation of HIPP cells may occur secondary to an overall evaluation of stimulus salience level during training.

Most cholinergic input into the hippocampal formation arises from the medial septum[32] and is triggered through the amygdala during conditioning with elemental stimuli[4]. In fact, the distinct circuits that mediate background or foreground context representations in the hippocampus are linked to the activation of projections from the medial septum and toward the lateral septum[3, 5]. Our data suggest that this cholinergic circuitry involves an M1 receptor-mediated activation of HIPP cells to control background context fear by regulating granule cell excitability and the size of DG memory ensembles. However, additional mechanisms appear to act in the DG as HIPP-independent activation of basket cells can also reduce background context memory. Further, M1 receptors are also prominently expressed on granule cells, and muscarinic signaling can directly control the excitability and plasticity of these neurons[33–35].

Thus while cholinergic stimulation of the hippocampal principle cells may be required for context memory formation *per se*, the concomitant activation of M1 receptors on HIPP cells appear to devaluate the background context information during elemental conditioning. Indeed, while Y1 receptor blockage did not affect perforant path-induced granule cell activity in hippocampal slices in the absence of muscarinic stimulation, it could efficiently decrease its oxotremorine M induced inhibition.

This effect was mimicked by the pharmacogenetic inhibition of HIPP cells, suggesting that these interneurons by releasing NPY, mediate the cholinergic inhibition of DG granule cells. In addition to their direct activation via M1 receptors, HIPP cells may also be activated indirectly, e.g., via acetylcholine induced glutamate release from hilar astrocytes[36].

HIPP cells represent the major population of NPY[+] neurons in the dentate gyrus (Supplementary Fig. 1f) and NPY in turn serves as co-transmitter in the majority of HIPP cells, inhibiting granule cells via Y1 receptors[37]. NPY in the DG has recently been proposed to reduce the stability of CS-US associations during eyeblink conditioning[38], which is in line with a role of NPY in the adjustment of memory salience. We could not only show that NPY indeed controls background context salience during memory encoding, but also that NPY mRNA expression levels in HIPP cells are regulated over a considerable range through the transcription factor CREB. This is in line with CREB acting as a regulator of NPY expression in neuronal cells[39] and the activity-dependent regulation of NPY in the hilus, e.g. in response to stress[40, 41]. The specific induction of phospho-CREB[S133] after background context conditioning indicates its activity-dependent regulation in HIPP cells and suggests a role in the replenishment of NPY stores. However, CREB may play additional roles in HIPP cells by controlling their excitability and/or molecular processes during memory consolidation[42]. In fact, we found that viral expression of CREB can also elevate GAD67 mRNA levels in

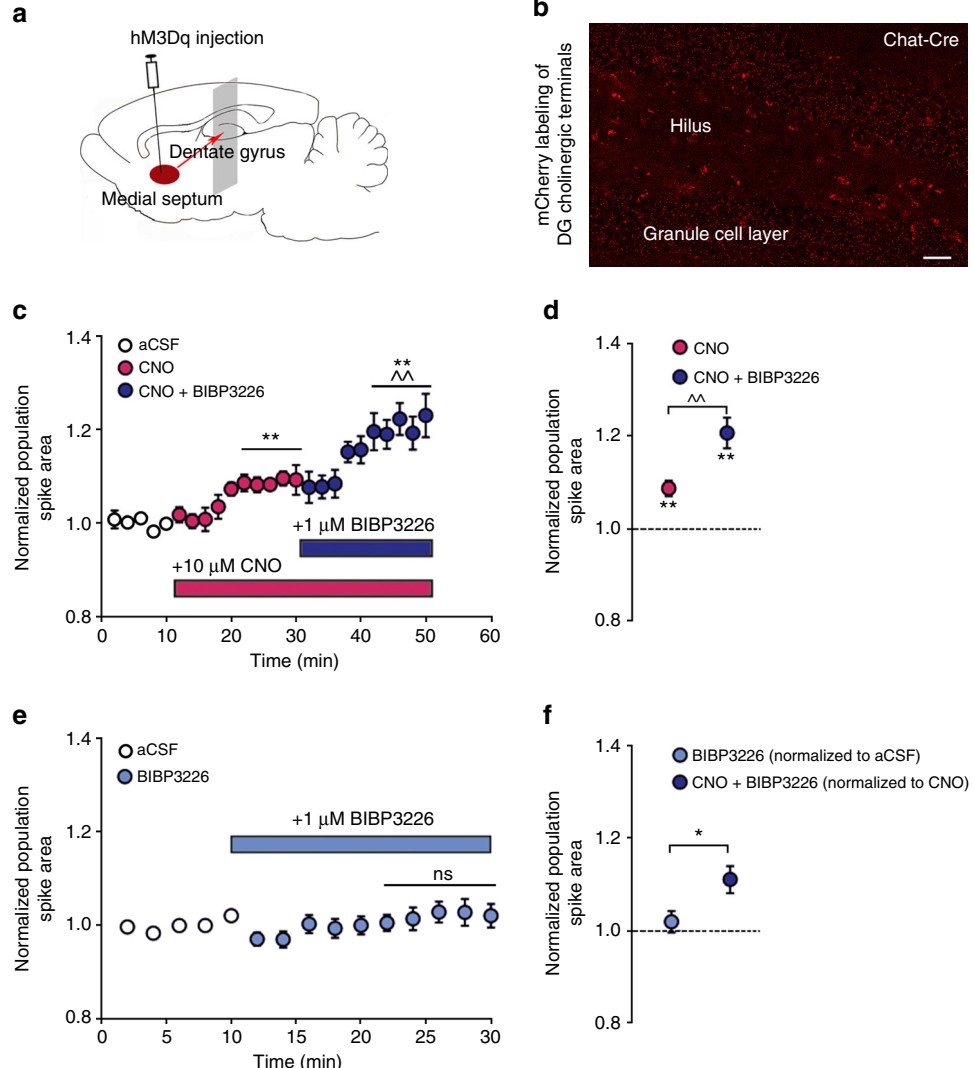

**Fig. 6** Facilitatory effect of endogenous ACh-release on the DG population spikes is augmented by Y1 receptor blockage. **a** Schematic of conditional hM3Dq viral injection into the medial septum of Chat-Cre driver mice. **b** A representative microscopic image shows the expression of the virally expressed mCherry tag in cholinergic terminals throughout the dentate gyrus. Scale bar, 25 μm. **c** The hippocampal cholinergic terminals expressing hM3Dq are activated by perfusion of hippocampal slices with CNO. Total of 10 μM CNO results in a moderate increase of the DG population spike area, which is further augmented by 1 μM BIBP3226 ($n = 8$ slices). For statistical analysis the last 10 min of each condition are averaged and compared, to ensure full availability or the bath-applied drugs. **d** Summary graph illustrating the change in average population spike area after CNO and CNO + BIBP3226 application relative to aCSF. **e** By contrast, BIBP3226 application in the absence of CNO does not affect the population spike area ($n = 16$ slices). **f** Comparison of normalized BIBP3226 effects on population spike area shows a significant change only after CNO application ($n = 16$ slices without CNO and $n = 8$ slices with CNO). Values are means ± s.e.m. Statistical analysis was done with Fisher's LSD following one-way repeated measures ANOVA (^^$P < 0.01$: CNO vs. CNO + BIBP3226; **$P < 0.01$; CNO or CNO + BIBP3226 vs. aCSF condition) in **c** and **d**, paired $t$-test in **e** and Student's unpaired $t$-test in **f** (*$P < 0.05$)

HIPP cells in a S133 phosphorylation dependent manner. On the other hand, the observed increase of SST mRNA after transduction of CREB[S133A] (similar to native CREB) may be indicative of a constitutive, phosphorylation independent regulation of SST expression[43].

Changes in GABAergic and/or somatostatinergic transmission in both NPY+ and NPY− HIPP cells may well have contributed to the behavioral effects of CREB- or activity-manipulation in our study. In this context it is interesting to note that only specific intervention with the muscarinergic and NPYergic transmission, but not the more general interference with HIPP cell functions through hM4Di or dominant negative CREB[S133A] also enhanced immediate context memory in our experiments. These differences may be related to additional cellular effects of the latter manipulations as well as to an involvement of both NPY+ and NPY−

subpopulations of HIPP cells and/or their interaction with other interneuron populations[17].

Nevertheless, our physiological data clearly indicate that NPY-mediated transmission is a major determinant of HIPP cell-mediated cholinergic inhibition. Their pharmacogenetic inhibition not only resembles the NPY receptor blockage in the DG, but the combined inhibition of HIPP cells (with hM4Di) and NPY receptor blockage did not have a larger effect on oxotremorine M induced inhibition than NPY receptor blockage alone. Notably, NPY blockage had no effect on the DG population spikes in the absence of cholinergic stimulation, in line with a recent report[20], but enhanced the excitatory effect of acetylcholine released from medial septum terminals in the dentate gyrus. HIPP cells thus appear to regulate the balance between excitatory and inhibitory effects of acetylcholine in the DG[36].

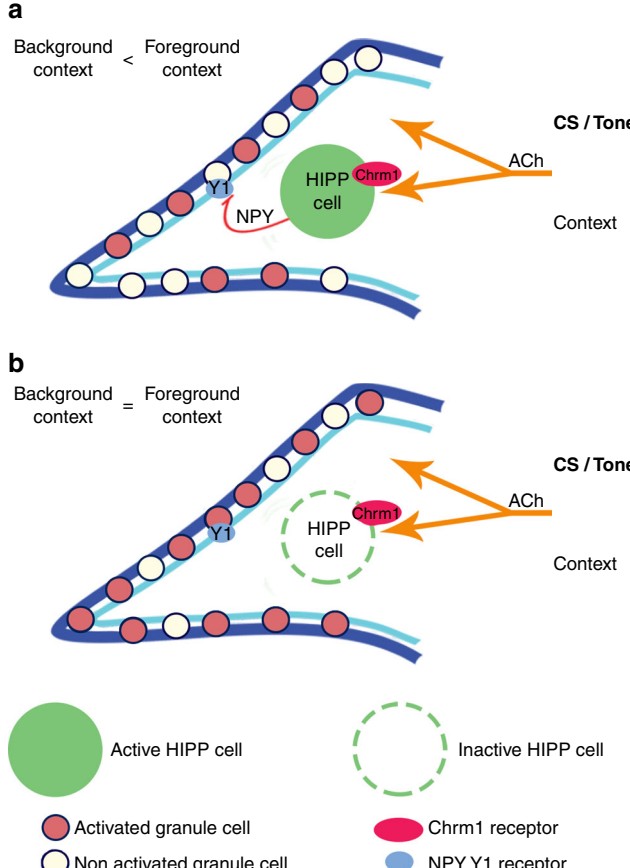

**Fig. 7** A local circuit model for the cholinergic modulation of background context salience in the dentate gyrus. **a** HIPP cells act as relay stations in the septo-hippocampal pathway that adjust context memory strength during fear learning. Acetylcholine released from medial septal afferences during memory acquisition activates both granule cells and HIPP cells in the dentate gyrus. The cholinergic stimulation of HIPP cells via muscarinic M1 receptors (Chrm1) triggers their NPY release, which in turn via Y1 receptors attenuates the excitation of granule cells. This feed forward inhibition is effective when conditioning occurs to an elemental cue (context in background) and restricts the number of granule cells that are recruited to the fear memory engram, resulting in reduced context salience. **b** Upon experimental silencing of HIPP cells and blockage of NPY-mediated inhibition the cue-induced devaluation of the background context fails and an increased number of granule cells are recruited to the context memory trace. Experience-dependent changes in HIPP cell function, e.g., via CREB-mediated transcription can adjust the strength of this local circuit and are critical for the function of the DG in adaptively encoding context salience

While our data suggest that this inhibitory HIPP cell circuit is utilized to control background context salience, others have also reported an increase of pure context memory upon pharmacogenetic blockage of HIPP cell activity[15] that was not evident in our experiments. Between lab variability in mouse phenotypes is frequently observed and may be attributed to minor differences in housing or training conditions[44, 45]. As HIPP cells control the size of granule cell ensembles by mediating the competition and lateral inhibition between training-activated and training-silent granule cells[15], it is plausible that this circuit may be more generally involved in the adjustment of context memory salience by acetylcholine.

Next to the DG, cholinergic mechanisms in CA3 have also been shown to facilitate the encoding of background context memories[46, 47]. In addition, Lovett–Barron and coworkers

recently reported an activation of SST+ interneurons in CA1, which appear to relay US-related information provided by the cholinergic projection from the medial septum during context conditioning[48]. We propose that cholinergic input from the medial septum stimulates the function of the DG to control pattern separation and context discrimination in parallel with generally enhancing context memory salience via areas CA1 and CA3. Thus dendrite-targeting local circuit neurons and NPY allow the hippocampus to adjust the relative weight of converging direct and indirect pathways from entorhinal cortex and thereby control synaptic plasticity in CA1[38].

This integration process, on one hand, depends on the level of acetylcholine release, which is less pronounced in the hippocampus during conditioning with an elemental CS than during pure context conditioning[49]. On the other hand, it depends on prior experiences and involves adaptive changes within the local circuitry and neuropeptide expression, as indicated by the effects of dominant negative CREB expression and associated regulation of NPY. In fact, dominant negative CREB expression in HIPP cells reduced oxotremorine M responsiveness albeit less efficiently than complete Y1 receptor blockage with BIBP3226. Ultimately the circuit is under influence of the lateral amygdala, which computes simple stimulus-shock associations and is required to restrict background contextual fear conditioning in the presence of a predictive elemental stimulus[4].

Together, our findings suggest that NPY+ HIPP cells are relay stations in this septo-hippocampal pathway adjusting context memory strength. Specifically, these interneurons allow the DG to discriminate full and partial recapitulation of the learned configuration during memory retrieval. The behavioral effects induced by a pharmacogenetic disruption of HIPP cells function or blockage of Y1 receptors are comparable in their magnitude to the difference in freezing levels induced by foreground or background context conditioning in this and other studies[3, 5]. Similar increases in background context fear memory have been observed in rodents after highly intensive auditory cued fear conditioning[50, 51] and after pre-exposure to juvenile stress[52]. Also, changes in neuropeptide Y expression levels in the dentate gyrus are observed after acute stress[40]. Thus, these changes may provide a neurochemical correlate for the adaptive and maladaptive adjustment of background context salience, such as observed following highly stressful experiences[53]. Indeed, different levels of NPY expression have been observed in the DG of rats sensitive or resilient to stress-induced behavioral pathology in a model of post-traumatic stress disorder[41]. It is thus likely that the HIPP cell circuit described here contributes to pathology development in PTSD, as an escalation of background context fear has been implicated in the occurrence of intrusive memories[54] and attenuated fear extinction[55] that are characteristic for this disorder.

## Methods

**Animals.** All mice were bred and raised in the animal facility at the Institute of Biology, Otto-von-Guericke University Magdeburg. The animals were kept in groups of 2–6 on an inverse 12 h light/dark cycle (lights on at 7 pm with a 30 min dawn phase) with food and water ad libitum. Animal housing and animal experiments were in accordance with the European regulations for animal experiments and approved by Landesverwaltungsamt Saxony-Anhalt (Permission Nr. 42502-2-1177-UniMD). The mouse lines used were: *NPY-GFP* (B6.FVB-Tg (Npy-hrGFP)1Lowl/J), *SST-Cre*$^{ERT2}$ (B6(Cg)-Sst$^{tm1(cre/ERT2)Zjh}$/J), *PV-Cre* (B6;129P2-Pvalb$^{tm1(cre)Arbr}$/J), *CamK2a-Cre* (Tg(Camk2a-cre)159Kln), *Chat-Cre* (B6;129S6-Chat$^{tm1(cre)Lowl}$/J and wild type C57BL/6BomTac (M&B Taconic, Germany). *SST-Cre*$^{ERT2}$ enabled us to manipulate HIPP cells without perturbing the NPY gene function and also allowed us to specifically address CREB activity and mRNA expression profiles in NPY+ cells of the hilus. A high degree of co-expression of NPY and SST in the hilus of our experimental animals and the corresponding targeting of HIPP cells by both *SST-Cre*$^{ERT2}$ and *NPY-GFP* transgenes were confirmed immunohistochemically (Supplementary Fig. 1f, g).

Tail biopsies were collected during weaning and mutant genotypes were determined with allele-specific polymerase chain reaction using the supplier's

recommended protocols. All experimental mutant mice were heterozygotes (Chat-Cre: homozygotes) and littermates, randomly assigned to experimental and control groups, and were tested in parallel.

**Behavioral experiments**. Mice were single-caged 3–4 days before the start of the experiments. Behavioral experiments were conducted during the dark phase, between 10 am and 4 pm. Before testing, animals were handled for 2 days by an experimenter who was blind with respect to treatment. Open field and elevated plus maze tests were performed as described previously[52], using low light conditions (10 lux). To address the role of HIPP cells in the adjustment of background context salience we optimized our training/testing protocol such that the fear response after foreground conditioning is maximal and not different from pure context conditioning (Fig. 1b).

Fear conditioning was conducted[50, 51] in a sound-isolated conditioning chamber containing a $16 \times 32 \times 20$ cm acrylic glass arena fitted with a grid floor for delivery of foot shocks, a loudspeaker and a ventilator (background noise 70 dB SLP, light intensity < 10 lux; TSE Systems, Bad Homburg, Germany). Before conditioning, mice were habituated individually to the conditioning chamber for 6 min twice a day for 2 days to provide a stable pre-training representation of contextual features without inducing latent inhibition[50]. On the third day different conditioning protocols were applied always beginning with 2 min of context exposure. For background context conditioning, after the initial 2 min, mice were exposed to three conditional stimuli (CS: 10 kHz tone for 10 s, 80 dB), each co-terminating with an unconditional stimulus (US foot shock: 0.4 mA for 1 s), with inter-stimulus intervals (ISI) of 20 s. For foreground context conditioning, we adapted an unpaired training protocol[49]. Mice received each three US foot shocks (0.4 mA for 1 s) and three tones (10 kHz tone for 10 s, 80 dB) without temporal coincidence and with variable ISIs ranging from 20 to 40 s. In pure context conditioning, mice received three un-signaled US foot shocks (0.4 mA for 1 s) only. Mice were returned to their home cages 2 min after the last stimulus. For a schematic overview of fear training paradigms see Fig. 1a.

For testing contextual fear memory, mice were individually placed in the training context for a period of 6 min, 24 h after training. In a second test session, 48 h after training, fear memory to the CS was assessed in a clean standard cage with bedding material serving as the neutral context. Animals after an initial 2 min habituation phase were exposed to a set of 4 CS (10 s, with each 20 s ISI). The animals' freezing behavior was assessed online via a photobeam detection system that detected immobility periods >1 s. This automated detection of immobility reliably reflects observer-rated freezing measurement[50, 51]. The freezing score was calculated as percentage of total time spent freezing during context exposure and CS presentations.

**Adeno-associated viruses**. AAV hSyn-DIO-hM4Di-mCherry, AAV hSyn-DIO-hM3Dq-mCherry and AAV hSyn-DIO-mCherry were obtained from the Vector Core Facility of University of North Carolina (Chapel Hill, NC, USA).

**Conditional lentiviral vector**. Lentiviral vectors were based on the vector backbone pLL3.7[56]. First, a dsRed cassette from MSCV-RmFF was used to replace GFP in MSCV-FlipFF[57], which harbors an empty miR30-FF cassette for the conditional expression of shRNA constructs. The combined dsRed/miR30FF cassette, including a loxP5171 site upstream and inverted loxP5171 and loxP2272 sites downstream, and an additional upstream loxP2272 site were excised with ApaI and BamHI. Both fragments were inserted to pLL3.7, thereby replacing the U6 and conditional CMV$_{\mathrm{IE}}$ promoters as well as the GFP cassette of pLL3.7. Finally, a CMV promoter from vector pHA-CMV was inserted yielding pLL-dfRmFF.

**Dominant-negative CREB**. Expression cassettes for native human CREB and CREB$^{S133A}$ were amplified from commercially available vectors and cloned into pCMV-HA (all Clontech, Saint-Germain-en-Laye, France) using EcoRI and NotI restriction enzymes. The so derived HA-CREB and HA-CREB$^{S133A}$ constructs and the isolated HA tag were transferred into pLL-dfRmFF in an inverted orientation, replacing mir30 and DsRed elements.

**Chrm1 knockdown**. The full-length Chrm1 coding sequence was amplified from pYX-Chrm1 (ImaGenes) using primers 5′-TAGGAACACCTCAGTGCCCC CTGC-3′ and 5′-TATTAGCATTGCTCGAGGCGGGAGGGGGTGC-3′ and inserted in frame to vector pCMV-HA (Clontech) using EcoRI and XhoI restriction sites. Five plasmids containing shChrm1 knockdown constructs in vector pLKO.1 (Sigma-Aldrich, Seelze, Germany) were tested in cell culture for their efficiency to knockdown the co-expressed HA-Chrm1 mRNA.

Authenticated NIH3T3 and HEK293T cells were directly obtained from the German Collection of Microorganisms and Cell Cultures, and regularly tested for mycoplasma contamination in our lab. NIH3T3 cells were cultured in Dulbeccos Modified Eagles Medium (DMEM) supplemented with 10% fetal bovine serum (FBS, both Life Technologies, Darmstadt, Germany) to 80% confluence. Cells were co-transfected with HA-Chrm1 and shLuc or shChrm1 plasmids using Lipofectamine 2000 (Life Technologies, Darmstadt, Germany), according to the manufacturer's protocol. Transfected cells were lysed in 1% dodecyl maltoside,

1% NP-40, 0.5% DOC, 50 mM Tris, (pH 7.5), 150 mM NaCl, 1 mM AEBSF, 1 μM Pepstatin and PIC tablet (Life Technologies, Darmstadt, Germany), separated by SDS–PAGE and transferred to PVDF membranes (Merck Millipore, Darmstadt, Germany). Membranes were probed with anti-HA (Merck Millipore, Darmstadt, Germany) and anti-tubulin (Sigma-Aldrich, Seelze, Germany) primary antibodies and fluorescence-labeled secondary antibodies (LI-COR, Germany). Protein bands were visualized using the Odyssey® imaging system (LI-COR, Germany), quantified in ImageJ and HA-Chrm1 amounts were normalized against tubulin levels.

Based on the observed knockdown efficiency (Supplementary Fig. 9c) an shRNA construct harboring the sequence 5′- CCGGCATGAACCTCTAT ACCACATACTGGAGTATGTGGTATAGAGGTTCATGTTTTT-3′ was selected for *in vivo* experiments. This sequence targets bp850–bp871 of Chrm1 transcript variant 1 (NM_001112697.1) and bp758–bp779 of Chrm1 transcript variant 2 (NM_007698.3). A suitable 97-mer oligonucleotide for expression via miR30[58] was designed using the Cold Spring Harbor Laboratory website (katahdin.cshl.org). Oligomers (Invitrogen, Karlsruhe, Germany) were amplified using the primers miR30 5′-GATGGCTGCTGGAGAAGGTATATTGCTGTTGACAGTGAGCG-3′ and miR30rev 5′-GTCTAGAGGAATTCCGAGGCAGTAGGCA-3′ to add XhoI and EcoRI restriction sites and subsequently cloned into pLL-dfRmFF using these sites. pLL-dfRmFF without insertion was used for the generation of control viral particles.

**Generation of lentiviral particles**. Each $10^6$ HEK293T cells were grown in DMEM with 10% FCS and each 1% Penicillin and Streptomycin (Life Technologies, Darmstadt, Germany) at 37 °C, 5% CO$_2$. On the day of transfection, fresh DMEM with 25 μM chloroquine was applied and cells further incubated for 2–6 h. When cells had grown to 80% confluence, transfection was done using the calcium phosphate method, with a final concentration of 0.125 M CaCl$_2$, 3 μg of pMD2. G, 5 μg of pRSV-Rev, 5 μg of pMDLg/pRRE and 10 μg of the respective pLL transfer vector. After overnight incubation, cells were further grown with fresh DMEM, 10% FCS and each 1% Penicillin and Streptomycin. Supernatant was collected on the following 2 days, filtered with a 0.45 μm syringe filter (Corning, Germany) and concentrated by ultracentrifugation for 2 h at 50.000 g (MLA-55, Beckman Coulter). Pellets were re-suspended in PBS and stored at −80 °C. Viral titer was determined using the QuickTiter™ Lentivirus Titer Kit (Cell Biolabs, USA) according to the manufacturer's instructions. Experimental vectors (HA-CREB, HA-CREB$^{S133A}$, shChrm1) were always prepared in parallel with the corresponding control vectors.

**Virus application**. Mice were anesthetized using 50 mg kg$^{-1}$ (i.p.), sodium pentobarbital and placed on a stereotaxic frame (World Precision Instruments, Berlin, Germany). After craniotomy, 33G injection needles attached to 10 μl NanoFil microsyringes (World Precision Instruments, Berlin, Germany) were lowered into the dorsal hilus anterioposterior (AP): −1.94 mm, mediolateral (ML): ±1.3 mm from Bregma and dorsoventral (DV): −1.7 mm from brain surface. Viral vectors ($10^9$ particles μl$^{-1}$) were injected bilaterally at flow rate of 0.1 μl per min via a digital microsyringe pump (World Precision Instruments, Berlin, Germany). Each hemisphere received 1 μl of virus solution. For medial septum injections viral vectors were injected unilaterally according to AP:+0.9 mm, ML: −0.6 mm from Bregma and DV: −4.7 mm from brain surface at an angle of 5°. Each mouse received 250 mg kg$^{-1}$ Novalgin® in 0.9% saline subcutaneously for post-operative analgesia and was allowed to recover from anesthesia under an infrared lamp. Behavioral experiments were performed 1 week after injection. To ensure efficient expression at cholinergic terminals, Chat-Cre mice were left for 8–10 weeks before slice preparation. SST-Cre$^{ERT2}$ mice received three daily 100 μl injections of tamoxifen (2, 4, and 8 mg, i.p.) to induce CRE activity; tamoxifen was first dissolved in 96% ethanol and then diluted to its final concentration in corn oil (Sigma-Aldrich, Seelze, Germany). This induction protocol was established empirically in our lab to ensure efficient activation of CRE recombinase in hilar NPY$^+$/SST$^+$ cells. In fact our tamoxifen treatment protocol proved superior to the initially reported method[59] and with a minimal adverse side effects (Supplementary Fig. 1e, g). hM4Di and hM3Dq receptors were activated 1 h before the fear memory training via i.p. injection of 10 mg kg$^{-1}$ clozapine-N-oxide (Enzo Life Sciences, Germany) in physiological saline[60].

Expression of viral constructs was verified histologically after completion of behavioral experiments. Firstly, to confirm the efficiency of viral vectors, cells expressing AAV or lentiviral control constructs were quantified on six alternating 30 μm sections per animal, from Bregma AP: −1.58 to AP: −2.18 mm, hence covering >60% of the rostro-caudal extension of the dorsal dentate gyrus. Positive cells were counted between the medial pole of the hilus and the beginning of area CA4 ($N = 6$ per construct) and related to corresponding area of the section. This quantification revealed a density of $0.683 \pm 0.067$ cells per 1000 μm$^2$ for AAV and $0.596 \pm 0.021$ cells per 1000 μm$^2$ for lentivirus indicating equally efficient transduction of hilar interneurons by both viral vectors. Immunohistochemical double labeling revealed that >95% of NPY$^+$ cells in the evaluated region were labeled with the virally transduced markers. For assessment of behavioral effects, only mice with accurate bilateral marker expression throughout the same region were considered.

**Pharmacological Testing**. Bilateral 26G guide cannulas (C235GS-5-2.0/SPC, Plastic One Inc. Roanoke, VA, USA) were stably implanted into the dorsal DG under stereotaxic guidance, as described previously[61] with the coordinates AP: −1.94 mm, ML:±1 mm from Bregma and DV: −1.4 mm below pedestal from the brain surface. For intracerebroventricular injections, a unilateral guide cannula was implanted above the left lateral ventricle; AP: −0.1 mm, ML: −1.0 mm from Bregma and DV: −1.1 mm from the brain surface.

After implantation, the guide cannulas were closed with dummy cannulas and covered with a dust cap. Mice were allowed to recover from surgery for 7 days before testing. For drug injection, mice were briefly anaesthetized with isoflurane and a 33G injection cannula was inserted, extending 0.5 mm beyond the guide cannula. After 5 min, mice received intra-hippocampal infusions of 1 μl per hemisphere of the antagonists or corresponding vehicle at a rate of 0.5 μl per min. The selective Y1 receptor antagonist BIBP3226 (((R)-N2-(diphenylacetyl)-N-[(4-hydroxyphenyl)methyl]-argininamide); Tocris Bioscience, Ellisville, Missouri, USA) was applied at 1.5 pmol μl⁻¹ in 0.9% saline, 1% dimethyl sulfoxide (DMSO), 45 min before fear memory training or fear memory retrieval. For i.c.v. injections, BIB3226 was applied at 30 pmol per 2 μl[62]. Scopolamine hydrobromide S1845 (Sigma-Aldrich, Seelze, Germany) was injected at doses of 2.5 μg μl⁻¹or 5 μg μl⁻¹ in 0.9% saline, 15 min before conditioning or testing in the elevated plus maze. Cannula placement was verified histologically after the completion of the behavioral experiments and mice with correct cannula placement were included in final analysis.

**RNA isolation and quantitative PCR**. Laser microdissection (P.A.L.M MicroBeam, Carl Zeiss, Jena, Germany) of the dorsal hilus from freshly frozen tissue was done as described earlier[50, 63]. The dorsal hilar tissue was collected bilaterally from 8 to 10 sections per animal (AP: −1.34 mm to AP: −2.54 mm) after injection of dominant negative CREB$^{S133A}$ or native CREB (Supplementary Fig. 4d). For mRNA profiles, GFP⁺ fluorescent cells were collected from paraformaldehyde (PFA) fixed tissue by laser microdissection. NPY-GFP transgenic mice received an overdose of sodium pentobarbital (80 mg kg⁻¹, i.p) and were perfused transcardialy with ice-cold RNase free Tyrode's buffer followed by 4% PFA in phosphate buffered saline (PBS). Brains were post-fixed in the same solution followed by cryoprotection in 30% sucrose in PBS at 4 °C. After 2–3 days brains were snap frozen in methylbutane cooled with liquid nitrogen, and 14-μm thick coronal sections were prepared on a cryostat and mounted on pLL-coated-RNase free membrane slides. GFP⁺ from the hilus and cornu ammonis were captured before the remaining hilar tissue was isolated (Supplementary Fig. 6a). Lysis of samples and total RNA isolation was performed via RNeasy® FFPE kit (Qiagen, Hilden, Germany), according to the manufacturer's instructions.

For quantitative PCR, TaqMan® reagents and predesigned FAM-labeled assays for NPY (Mm00445771_m1), SST (Mm00436671_m1), GAD67 (Mm00725661_s1), GAD65 (Mm00484623_m1), Chrm1 (Mm01231010_m1), Chrm2 (Mm01167087_m1), Chrm3 (Mm01338410_m1), Chrm4 (Mm00432514_s1), Drd2 (Mm00438545_m1), Drd3 (Mm00432889_m1), Adr1d (Mm01328600_m1), 5Ht2c (Mm00664865_m1), Grik1 (Mm01150783_m1), Grik2 (Mm01181235_m1), and PGK (Mm00435617_m1) were used in duplex assays with VIC-labeled assay for glyceraldehyde 3-phosphate dehydrogenase (GAPDH) serving as internal control.

**Immunohistochemistry**. SST, pCREB$^{S133}$, and HA were detected according to a previously described immunohistochemical method[64, 65]. Anti HA-tag (Cell Signaling) at a dilution of 1:300, anti-pCREB$^{S133}$ (Cell Signaling, Frankfurt am Main, Germany) at 1:250, anti-NPY (Abcam, Cambridge, UK) at 1:700 and anti-SST (Santa Cruz Biotechnology, Heidelberg, Germany) at 1:250 were used as primary antibodies and detected with Alexa-Fluor488 and Alexa-Fluor555-coupled secondary antibodies (Life Technologies, Darmstadt, Germany), both at 1:1000. Nuclei were visualized with DAPI (300 nM). cFos, cholinergic fibers and M1 receptors were stained according to protocols described previously[65]. Anti-cFos (Cell Signaling #2250) was used at 1:1000, anti-ChAT (Novus biological, USA) at 1:100 and anti-M1 (Sigma-Aldrich, Seelze, Germany) at 1:250 dilutions. As secondary antibody, biotinylated anti-goat (1:200; Vector Laboratories, USA) was used and visualized through conjugation with streptavidin Cy2 or Cy3 (both 1:1000; Jackson ImmunoResearch Labs, UK). Immunostainings were examined using a DMI6000 epifluorescence microscope and a TCS SP2 or a SP8 confocal microscope (all Leica, Germany). To avoid bleaching through the individual color channels (DAPI, GFP, Cy2, Cy3) were scanned sequentially.

For quantification of cFos⁺ (Fig. 1h), CREB⁺ and NPY⁺ cells (Fig. 2g), the brains of conditioned mice were serially cut into 30-μm thick coronal sections. Six sections per animal were stained and evaluated at intervals of 90 μm starting from the rostral pole of the DG. cFos⁺ cells were counted in the granule cell layer of the dorsal DG, CREB⁺ and NPY⁺ cells in the hilus and CA1 *stratum oriens*. Area sizes were measured using the inbuilt LAS-AF Software and cell densities were expressed as number of positive cells per mm² or, for NPY⁺/CREB⁺ cells, as proportion of NPY⁺ cells. Mean cell densities of six slices for each individual animal were calculated and used for statistical comparison between groups.

**Field potential recordings**. To determine the effect of HIPP cell manipulation on DG activity we examined population spikes induced by perforant pathway stimulation. Adult male mice (wild type C57BL/6BomTac, SST-CRE$^{ERT2}$ mice or Chat-CRE driver mice) were deeply anesthetized with isoflurane and decapitated. Brains were rapidly (~ 30–60 s) removed and placed in cold (4–8 °C) carbogenated (5% CO₂/95% O₂) artificial cerebrospinal fluid (aCSF) containing (in mM) 129 NaCl, 21 NaHCO₃, 3 KCl, 1.6 CaCl₂, 1.8 MgSO₄, 1.25 NaH₂PO₄, and 10 glucose. Dorsal hippocampal transverse slices (400 μm) were obtained from the septal pole by cutting parasagittal slices at an angle of about 12° using an angled platform. Three to four most dorsal slices were transferred to an interface chamber perfused with aCSF at 32 ± 0.5 °C (flow rate: 2.5 ± 0.2 ml per min, pH 7.4, osmolarity ~ 300 mosmol kg⁻¹). Slices were incubated at least for ~ 1 h before starting recordings. Extracellular field recordings were obtained with a glass electrode filled with aCSF (~ 1 MΩ). For DG electrophysiology, the recording electrode was placed at 70–100 μm depth on the granule cell layer. Stimulation presumably activating both medial and lateral fibers in the perforant pathway was performed using a bipolar tungsten wire electrode with exposed tips of ~ 20 μm and tip separations of ~ 75 μm (electrode resistance in aCSF: ~ 0.1 MΩ). Before obtaining an input–output (I/O) curve, 20–30 min of baseline responses were recorded (0.033 Hz, pulse duration: 100 μs). Seven intensities ranging from 10 to 150 μA were used to obtain an I/O curve. The stimulus intensity that resulted in ~ 70% of the maximum population spike (PS) amplitude was further used for baseline excitability measurements. To analyze the effects of the selective Y1 receptor antagonist BIBP3226 (1 μM), slices were obtained from wild type C57BL/6BomTac mice. The HIPP cells were silenced via CNO (10 μM) using slices obtained from SST-CRE$^{ERT2}$ mice. After 20 min of baseline recordings (one stimulus every 2 min) either the selective Y1 receptor antagonist BIBP3226 (1 μM) or CNO (10 μM) was added to the aCSF and remained in the perfusion solution for the rest of the experiment. Following 20 min of only BIBP3226 or CNO application, oxotremorine M (oxo, 1 μM) was added to the solution and PS responses were measured for another set of 20 min. In the second set of experiments key findings were confirmed using carbachol (CCh, 10 μM) as muscarinergic agonist. Control slices were measured in a similar manner without BIBP3226 or CNO application. A similar protocol was used to analyze the effects of combined application of BIBP3226 and CNO vs. only BIBP3226 in slices obtained from SST-CRE$^{ERT2}$ mice. To measure muscarinergic depression of the PS responses in mice expressing either native CREB or CREB$^{S133A}$, low (0.1 μM) and high (1.0 μM) concentrations of oxo were used. Similarly, after 20 min of baseline recordings (one stimulus every 2 min), BIBP3226 was added to the aCSF and remained in the perfusion solution for the rest of the experiment, which was then followed by application of oxo for 20 min. Control slices were measured in a similar manner without BIBP3226. To investigate the effects of endogenous ACh release on perforant path stimulation induced PS in slices, Chat-Cre driver mice were expressing hM3Dq in septo-hippocampal projection neurons were used. Expression of hM3Dq in terminals of these cells was confirmed in the dorsal DG (Fig. 6b). In parasagittal dorsal hippocampal slices, 20 min after baseline recordings (one stimulus every 1 min), CNO (10 μM) was added to the aCSF to stimulate the cholinergic fibers innervating the dorsal hippocampus. CNO remained in the perfusion solution for the rest of the experiment. After 20 min of CNO perfusion, BIBP3226 was added and responses were recorded for another 20 min. Signals were pre-amplified using a custom-made amplifier and low-pass filtered at 3 kHz. Signals were sampled at a frequency of 10 kHz and stored on a computer hard disc for off-line analysis. Data were analyzed offline using self-written MATLAB-based analysis tools (MathWorks, Natick, MA, USA). The PS area was calculated by integrating the area above the curve (ms.mV).

**Patch clamp recordings**. Spiking properties of HIPP cells were determined to examine their activity modulation through M1 receptors. Mice were decapitated after deep anesthesia with isoflurane (1-Chloro-2,2,2-trifluoroethyl-difluor-omethylether). The protocol is followed as described earlier[66]. A block of brain tissue containing the hippocampus was rapidly removed and placed in ice-cold oxygenated physiological saline containing (in mM): KCl 2.4; MgSO₄ 10; CaCl₂ 0.5; NaHCO₃ 24; NaH₂PO₄ 1.25; glucose 10; sucrose 195, bubbled with 95% O₂/5% CO₂. 300-μm thick transversal slices were cut via Microslicer™ (Ted Pella, USA), and incubated in a solution of the following composition (in mM): NaCl 125; KCl 2.5; NaH₂PO₄ 1.25; NaHCO₃ 24; MgSO₄ 3.8; CaCl₂ 0.2; glucose 10; bubbled with 95% O₂/5% CO₂ to a final pH of 7.3. Slices were allowed to recover at 34 °C for 20 min and maintained for up to 5 h at 20 °C. Single slices were transferred to a submersion chamber and perfused continuously at a rate of approximately 2 ml per min at 30 °C ± 1 °C with ACSF (in mM): NaCl 125; KCl 2.5; NaH₂PO₄ 1.25; MgSO₄ 2; CaCl₂ 2; NaHCO₃ 22; glucose 10, bubbled with 95%O₂/5% CO₂. 10 μM pirenzipine (Sigma-Aldrich, Seelze, Germany), 10 μM 6,7-dinitroquinoxaline-2,3-dione (DNQX), 50 μM DL-2-amino-5-phosphono-pentanoic-acid (AP5) and 10 μM bicuculline methiodide (all Tocris Bioscience, Ellisville, Missouri, USA) were added to the ACSF to continuously block synaptic transmission. 10 μM Oxotremorine M (Tocris Bioscience, Ellisville, Missouri, USA) was applied only once for 3–4 min to each slice.

Whole cell patch-clamp recordings were performed using a patch-clamp amplifier (EPC-9, Heka Elektronik, USA) under visual control of differential interference contrast infrared video microscopy (S/W-camera CF8/1, Kappa, Germany). A monochromator (Polychrome II, Till Photonics) connected to an

epifluorescence system and a 40*/0.80 water immersion lens was used to identify GFP[+] interneurons. Patch pipettes were pulled from borosilicate glass (GC150T-10, Clark Electromedical Instruments) to resistances around 3 MΩ and were filled with (in mM): K gluconate 95; $K_3$ citrate 20; NaCl 10; HEPES 10; $MgCl_2$ 1; $CaCl_2$ 0.1; EGTA 5; MgATP 3; NaGTP 0.5 (pH 7.25 with KOH). A liquid junction potential of 10 mV was corrected. Hyperpolarizing current pulses had an intensity of −30 pA and duration of 500 ms. Depolarizing current pulses were adjusted to elicit 1–4 spikes (+100 to +250 pA, 500 ms).

**Data analysis**. All values are given in mean ± s.e.m. Normality of data was tested with Shapiro-Wilk's test. For multiple comparisons one-way analysis of variance (ANOVA) was performed, followed by Fisher's protected least significant difference test (LSD) for post-hoc comparison. Either Student's two-tailed $t$-test or Mann–Whitney U test was used whenever only two groups were compared. Post-hoc analysis using the program g*power v3.1 confirmed that critical $t$ and $F$ values were reached in each case of reported significance. For field electrophysiology, when appropriate, either one-way repeated ANOVA, paired $t$-test or Wilcoxon signed rank test were used. A two-way ANOVA was used for assessing in field electrophysiology data the interactions of viral manipulation of CREB activation and BIBP3226 treatment. For patch clamp electrophysiology, data were analyzed non-parametrically using Wilcoxon signed rank test. Differences were considered statistically significant, if $P < 0.05$.

**Data availability**. An overview of statistical comparisons is given in Supplementary Table 1. The data that support the finding of this study are available from the corresponding author upon reasonable request.

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

## Acknowledgements

We are grateful to A. Kröber, F. Webers and S. Stork for expert technical assistance and to T. Porcuzek for excellent animal care, as well as Dr. H. Beck and K. Piwelleck for help with Chat-Cre driver mice. This work was supported by the German Research Foundation (CRC779/B05 to H.S. and O.S., CRC779/B06 to V.L., STO488/6 and CRC854/B10 to OS) and the federal state of Saxony-Anhalt and the "European Regional Development Fund" (ERDF 2007–2013), Vorhaben: Centre for Behavioural Brain Sciences (CBBS).

## Author contributions

S.A.R., A.A. and O.S. contributed to the study design and performed in vivo pharmacology, viral manipulations, gene expression, and behavior analysis. G.C. performed field recordings. B.M. developed lentiviral constructs. Y.E.D. performed Western blot analyses. S.L. performed scopolamine pharmacology, SST/NPY immunohistochemistry, and contributed to the development of M1 knockdown. S.M. and V.L. conducted the electrophysiological analysis of HIPP neurons. R.H. and B.S. contributed in microscopic image analysis. N.F. and H.S. contributed the anatomical analysis of cholinergic fibers and M1 receptor expression in NPY-GFP mice. S.A.R., A.A., G.C. and O.S. wrote the paper. All authors discussed and commented on the paper.

## Additional information

**Competing interests:** The authors declare no competing financial interests.

