## [Peer Review File · Nature Communications]

Reviewers' Comments:

Reviewer #1 (Remarks to the Author):

The manuscript of Raza et al describes a series of experiments that argue SST+ HIPP neurons in the dentate hilus are recruited via cholinergic input from the medial septum during background fear conditioning, resulting in NPY release onto DG granule cells and a decrease in contextual fear. The authors utilize an SST-Cre line to target the hilar HIPP cells and show a consistent behavioral phenotype across a series of cell-type specific manipulations (DREADD inhibition, dnCREB, M1R knockdowns) and local pharmacology (Y1R blockade). Overall the paper is well written and the data strongly support a novel role for this local circuit in the dentate, however there are several key points that need to be addressed by the authors to solidify their conclusions.

1. When interpreting the SST+ neuron manipulation data one main issue I have is determining the specificity of the affected population in terms of NPY. The authors state that 95% of NPY+ neurons in the hilus express their Cre-dependent AAV in the SST-Cre mice, suggesting they are targeting the majority of these neurons. However on p.7 they state only about 2/3rds of the SST+ neurons express NPY, thus a large population of the hilar neurons they are manipulating are not involved in NPY signaling. Further, the images include in Figs. 1e and 5d both demonstrate a high fraction of virally expressing neurons are NPY-. Given that the main conclusion of the paper is that NPY signaling mediates the decrease in background context fear, the contribution of these non-NPY SST+ neurons must be addressed. While the authors reference the recent Stefanelli et al paper demonstrating that GABA-mediated inhibition of GCs from SST+ INs limits the size of the active ensemble a more detailed discussion of the current data set (and it's above limitations) in the context of the previous work/GABA signaling would improve the paper.

2. The Stefanelli et al paper demonstrated that DREADD inhibition of SST+ neurons increased foreground context fear (Figure 6B) using a very similar 3-shock protocol. The discrepancy between that data and the current findings should be directly discussed.

3. The authors provide no direct evidence of DREADD mediated inhibition of HIPP neurons. In Figure 4d they do provide some indirect evidence, as CNO activation in vitro mimics the Y1R blockade experiment, however the data would be strengthened by demonstrating in vivo effects of DREADD-mediated inhibition of the SST+ neurons. This could be based on IEG expression or pCREB activation in these neurons following background conditioning (as in figure 2g) in the presence or absence of CNO administration or at the very least evidence of increased cFos expression in GCs as in Figure 6c of the Stefanelli paper.

4. In the dnCREB experiment the authors describe three genes as putative CREB targets, however present three unique patterns of modulation following either CREB inhibition or overexpression. It is unclear to me why the SST and GAD data is included. If presented this result needs to be addressed in the discussion. Further the authors should consider the possibility that the inconsistent results are related to the expression of the virus in NPY- cells.

5. In figure S4 the authors demonstrate that a 'higher, anxiogenic dose' of BIBP3226 delivered icv does not affect behavior and does not replicate the DG infusion data. I do not

understand how this data supports their claim that Y1 receptors are required for the decrease in context fear in the background condition. Do the authors assume this 20x larger dose leads to an insufficient blockade of Y1Rs in the DG? This should be made clear.

6. While both the physiological and behavioral data is solid, a logical connection between these findings is not made. Inhibition of hilar SST+ INs limits the CCh induced depression of the DG population spike. I realize interpreting these data is difficult in light of the complicated local circuit, however an effort should be made to relate how a loss of NPY-mediated modulation of this depression would result in enhanced background conditioning.

Reviewer #2 (Remarks to the Author):

In this manuscript the authors study the role of hilar interneurons that are associated with the perforant path (HIPP interneurons) in the dentate gyrus, in fear conditioning. Using transgenic mice they first show that silencing HIPP interneurons prior to learning using an inhibitory DREADD enhances memory of the background context – that is that associated with the conditioned stimulus, but has no effect on cued memory (tested in a different context). They then go on to show that this effect requires the activity of muscarinic M1 receptors and Y1 NPY receptors. Moreover, it appears that the effect of NPY is mediated by CREB mediated expression of NPY. Overall this is a nice piece of work that establishes the role of HIPP interneurons in the encoding of background context, and the data support their proposal that this encoding requires the activity of the neuropeptide NPY.

The role of the dentate gyrus in pattern separation is well established, and recent experiments using genetic manipulations are showing the key role it plays in context discrimination during fear learning. The involvement of the cholinergic system in this is also well known. Moreover, the role of hilar interneurons in spatial learning is reasonably well established. The new data here are showing the role of HIPP interneurons in context salience, and suggesting a role for CREB and Y1 receptors.

In this study they show that cholinergic modulation requires Y1 activity there is no indication how these two receptor systems interact, and what the impact of CREB mediated expression of NPY is. It is surprising that a peptide transmitter system is genetically upregulated and it interacts with the cholinergic system but one is left wondering what the impact of this is. Is there more NPY released? Is it only released following fear conditioning and CREB activation? How does NPY affect cholinergic activity.

Other concerns:

1. The impact of cholinergic activation on HIPP cell activity and its overall impact are only shown with bath application of agonists and antagonists. It is entirely unclear what the impact of synaptic stimulation of cholinergic afferents would be. This really needs to be shown as the tools for this are widely available.
2. When testing the effect on HIPP cells they use oxotremorine, as somewhat selective M1/M2 agonist and show its effect is blocked by pirenzepine. However, for the field potential data they switch to carbachol, which is much less selective, and no antagonist is shown. I suggest that both experiments should be shown using the same agonist and antagonist.

3. HIPP interneurons are SST positive, however, there is another population of PV-positive interneurons in the hilus. SST interneurons are known to innervate PV-interneurons. What then is the role of PV-interneurons?

Reviewer #3 (Remarks to the Author):

Overall

In this manuscript, the authors demonstrate that hilar perirhinal cortex-associated (HIPP) cells of the DG mediate the devaluation of background context memory during fear conditioning. The authors reduced NPY expression and blocked NPY-Y1 receptors in the DG to show this sensitivity. Moreover, they show that M1 muscarinic receptors mediate the cholinergic activation of HIPP cells and mediate HIPP cells' control of background context salience. I have concerns with the figures and statistical analysis. Some additional control experiments are needed.

Major Comments

Results

1. The authors should include all data, especially when conditioning was done without any CS presentation (page 5, line 104). The stats should not just be recorded. The data should be included at least in supplement. See also page 5, line 109; page 8, line 168; etc.
2. The authors should elaborate or hypothesize why both viral vectors increased SST expression (page 6, line 133).
3. The authors administer a BIBP3226 injection before retrieval (page 8, line 168). This type of experiment should be performed in all figures. To assess whether these cells are necessary for encoding and/or retrieval is essential. For example, in figure 1, CNO was given before acquisition. Why isn't the CNO additionally given in a separate cohort of mice before retrieval? These experiments should be included.
4. As an example, the authors state that *Chrml* is not observed in GFP- hilar neurons or GFP+ hippocampal neurons. The authors give the ANOVA, but not present the follow up p values for the t-tests or Fisher's LSD. This is consistently a problem throughout the paper. Moreover, in the aforementioned figure, if the GFP+ neurons are indeed significantly different from the other two groups, why is there only one asterisk in that panel in Figure 4A? Why is the hilar GFP+ v. hippocampal GFP+ difference not presented? I highly suggest that the authors create a table with all statistical analyses performed, p values, follow up tests, etc. As the manuscript is right now, the statistical analyses need major improvement. (please see page 9, line 192; Figure 4a, panel 2).
5. The authors should also consider order effects. I recommend performing at least Figure 1b and reversing the order of shock context and CS presentations (e.g. CS presentation on day 15 and shock context on day 16). It would further increase the validity of the increased fear expression in the background group.
6. Figure S4: The authors should more fully discuss the increased post-training freezing in the experimental group. This is problematic. Moreover, I recommend scoring the shock reactivity (distance traveled during the shock). Do the experimental mice respond differently to the shock?

7. Figure S6: Why is the 2.5 group should be included.

PM/analysis/This omitted from the E

8. Figure S6: Moreover, the authors write: “both doses further reduce background fear memory but not auditory cued fear memory compared to vehicle-injected controls.” In panel A, the 2.5 g/mg scopolamine is not. Something is incorrect in either the figure legend or in the stats presented in panel A.

g/mg

9. The authors should clearly explain the rationale of the experiments on page 9: the passage from oxotremorine to carbachole is not clear, they are both AchR agonist but they act on different receptors, also in the first case they investigate the spike frequency in the second case the population spike area. Moreover, they should indicate the exact values for Fisher's LSD one-way ANOVA, Wilcoxon signed rank test, and paired t-test.

Minor Comments

Introduction

1. The authors discuss the sufficiency of engrams in the DG for contextual memory, but completely omit the necessity work on DG engrams. The authors should include relevant literature on both necessity and sufficiency in the engram paragraph as well as in other relevant areas.

2. The authors state that the size of the activated DG granule cell ensembles correlates with contextual memory strength, but again omit the work performed on CA3. Their overview of engrams is overly simplistic and should consider the output of the DG to CA3 in their strength of a given memory discussion.

Methods

3. Page 16, line 366: there is a word missing before allowed. The sentence does not make sense as it.

4. Why are the mice single caged before the start of experiments? This is social isolation and is not appropriate for the experiments. The authors should perform all experiments in group housed mice for future experiments. I am not suggesting that all experiments be redone for this manuscript, just that they change this protocol for future experiments.

5. Why do the authors administer 2, 4, and 8 mg ip. These doses seem incredibly high. Why not administer 2-3 mg per day for 5 days to induce recombination? Was this previously published?

Figures

6. Why is each panel not referenced in the text? For example, Figure 1D-1E are not referenced. Each panel should be referenced.

7. Figure 2C: The authors state that the two experimental groups are different than the control group for SST. However, only 1 comparison line is shown. The authors should report both post-hoc tests and show two separate lines from control to experimental group as they did in the NPY group.

8. Figure 2: An experimental schematic should be included.

9. Figure 3: An experimental schematic should be included. Furthermore, Figure S4 and Figure 3 should be combined into 1 main figure.

10. Figure 5C and 5D: Again, these panels are not referenced.
11. Figure S1A: I suggest moving this schematic to Figure 1. It would be useful for readers.
12. Fig. 4C: The authors should add a graph showing the spike frequency (In the first big representative trace, the increase in the spike number is not so evident like in the single representative traces below). I suggest they can put together Fig 4C and S5D (panel 3).
13. Fig. S5D: The authors should specify the duration and intensity of the current pulses they used to elicit spike frequency.
14. Figure S5D: In the legend to panel D, there is a spelling error for pirenzepine.

RESPONSE LETTER

We are glad that all three reviewers have acknowledged the importance and originality of our study. This has encouraged us to systematically revise the manuscript and to include several new sets of data in order to address their points of criticism; we believe that we have been able to resolve all open points and that the changes made have indeed significantly improved our paper:

Reviewer #1 (Remarks to the Author):

The manuscript of Raza et al describes a series of experiments that argue SST+ HIPP neurons in the dentate hilus are recruited via cholinergic input from the medial septum during background fear conditioning, resulting in NPY release onto DG granule cells and a decrease in contextual fear. The authors utilize an SST-Cre line to target the hilar HIPP cells and show a consistent behavioral phenotype across a series of cell-type specific manipulations (DREADD inhibition, dnCREB, MIR knockdowns) and local pharmacology (Y1R blockade). Overall the paper is well written and the data strongly support a novel role for this local circuit in the dentate, however there are several key points that need to be addressed by the authors to solidify their conclusions.

Points:

1. When interpreting the SST+ neuron manipulation data one main issue I have is determining the specificity of the affected population in terms of NPY. The authors state that 95% of NPY+ neurons in the hilus express their Cre-dependent AAV in the SST-Cre mice, suggesting they are targeting the majority of these neurons. However on p.7 they state only about 2/3rds of the SST+ neurons express NPY, thus a large population of the hilar neurons they are manipulating are not involved in NPY signaling. Further, the images include in Figs. 1e and 5d both demonstrate a high fraction of virally expressing neurons are NPY-. Given that the main conclusion of the paper is that NPY signaling mediates the decrease in background context fear, the contribution of these non-NPY SST+ neurons must be addressed. While the authors reference the recent Stefanelli et al paper demonstrating that GABA-mediated inhibition of GCs from SST+ INs limits the size of the active ensemble a more detailed discussion of the current data set (and it's above limitations) in the context of the previous work/GABA signaling would improve the paper.

REPLY

Thank you very much for pointing this out. The expression levels of NPY in HIPP cells vary from cell to cell. With our rigorous evaluation of immunohistochemical stainings, we observed clear double labeling in approximately 2/3 of those cells.

GABA-mediated transmission is indeed highly relevant for the role of HIPP cells in DG information processing (Savanthrapadian *et al.*, 2014, J. Neurosci. 34, 8197– 8209), presumably in both NPY⁺ and NPY⁻ subpopulations including cells with NPY expression below the detection threshold. We discuss this now more explicitly in the discussion section on page 16 line 20 – page 17 line 3.

However, our data showing no effect of HIPP cell blockage or NPY-receptor blockage on population spike responses to PP stimulation in the absence of cholinergic stimulation

(Supplementary Fig. S7a) are in line with a recent report (Lee *et al.*, 2016, *Sci Rep.* doi: 10.1038/srep36885). Please see page 17 lines 4-13.

We have in the previous version of our manuscript shown largely similar effects of NPY receptor blockage by BIBP3226 and HIPP cell inactivation with hM4Di receptors after muscarinic stimulation. To this end, we have now added an experiment showing that combined inhibition of HIPP cells with hM4Di and NPY receptor blockage did not have a larger effect on oxotremorine M induced inhibition than NPY receptor blockage alone (Supplementary Fig. S7b). Together these data suggest that the muscarinic inhibition mechanism in the DG that controls background context salience may be largely mediated by NPYergic transmission (please see page 17 lines 4-13).

2. The Stefanelli *et al* paper demonstrated that DREADD inhibition of SST+ neurons increased foreground context fear (Figure 6B) using a very similar 3-shock protocol. The discrepancy between that data and the current findings should be directly discussed.

REPLY

Indeed, under our experimental conditions, we are not able to reproduce the ca 10% increase induced by HIPP cell inactivation on pure context memory as reported by Stefanelli *et al.*, 2016 (Neuron, 89,1074-1085), even when using the same mouse line as in their study (Sst^{tm2.1(cre)Zjh}/J) (Referee Figure 1). Thus although the freezing values of both studies are quite comparable, a difference is evident in the responsiveness of pure context to HIPP cell intervention and HIPP-mediated regulation. Even larger between-lab variability in mouse phenotypes has been observed previously and has been attributed to small differences in housing or training conditions. These have been and still are a matter of discussion (Crabbe *et al.*, 1999, *Science*, 284, 1670-1672; Richter *et al.*, 2009, *Nature methods*, 6, 257-261). We point this now on page 17 lines 14-19.

Referee Figure 1. Silencing of HIPP cells with conditional hM4Di viruses ($n=7$) in Sst^{tm2.1(cre)Zjh}/J mice by injection of CNO 1 h before pure context training does

not affect memory to the shock context compared to controls ($n=7$). The CS response is low as expected and without any difference between the groups.

Importantly for our study, to specifically address the role of HIPP cells in the adjustment of background context salience, we optimized our training/testing protocol such that the fear response after foreground conditioning is maximal and not different from pure context conditioning (now shown in Fig. 1b).

Moreover, we stress that HIPP neurons are likely not only involved in the discrimination of background and foreground context memory, but also in other aspects of context salience determination, particularly if involving ACh release in the hippocampus. Please see page 17 lines 19-22.

3. The authors provide no direct evidence of DREADD mediated inhibition of HIPP neurons. In Figure 4d they do provide some indirect evidence, as CNO activation in vitro mimics the Y1R blockade experiment, however the data would be strengthened by demonstrating in vivo effects of DREADD-mediated inhibition of the SST+ neurons. This could be based on IEG expression or pCREB activation in these neurons following background conditioning (as in figure 2g) in the presence or absence of CNO administration or at the very least evidence of increased cFos expression in GCs as in Figure 6c of the Stefanelli paper.

REPLY

We are thankful to the reviewer for raising this point and added an experiment to determine cFos immunoreactivity in the DG upon background context conditioning with HIPP cells inhibition. SST-CRE^{ERT2} mice were bilaterally injected with hM4Di viruses in the hilus and HIPP cells were silenced with CNO at the time of training. We found that silencing HIPP cells this way results in an enhanced induction of cFos expression in the DG, in accordance with the effect described previously by Stefanelli *et al.*, 2016, indicating a reduction of HIPP-cell mediated inhibition of granule cells and increase of ensemble size (Fig. 1h). Please see page 6 lines 14-16 and page 17 lines 19-22).

4. In the dnCREB experiment the authors describe three genes as putative CREB targets, however present three unique patterns of modulation following either CREB inhibition or overexpression. It is unclear to me why the SST and GAD data is included. If presented this result needs to be addressed in the discussion. Further the authors should consider the possibility that the inconsistent results are related to the expression of the virus in NPY- cells.

REPLY

SST and GAD67 as typical markers of HIPP cells were included in the mRNA analysis to control for the specificity of CREB-mediated NPY regulation. Their putative regulation in both NPY⁺ and NPY⁻ HIPP cells and implications for HIPP cell function are now better explained in the discussion on page 16 line 13 – page 17 line 3.

5. In figure S4 the authors demonstrate that a ‘higher, anxiogenic dose’ of BIBP3226 delivered icv does not affect behavior and does not replicate the DG infusion data. I do not understand how this data supports their claim that Y1 receptors are required for the decrease in context fear in the background condition. Do the authors assume this 20x larger dose leads to an insufficient blockade of Y1Rs in the DG? This should be made clear.

REPLY

In fact the penetration of drugs from CSF to brain parenchyma is very ineffective (Pardridge WM, 2011, Fluids Barriers CNS 8, 7.) and thus much higher concentrations are required to achieve biological effects. Our icv experiment was performed with a dose described earlier (Redrobe *et al.*, 2002, Neuropsychopharmacology. 26, 615-624), in order to demonstrate that reported anxiogenic effects of BIBP3226 (likely involving other brain regions like the amygdala) are not sufficient to induce a change in context memory salience as observed with our intra-hippocampal injections. We have pointed this out in more detail in the results section on page 9 lines 11-14 and the legend to Supplementary Fig. S5f.

6. While both the physiological and behavioral data is solid, a logical connection between these findings is not made. Inhibition of hilar SST+ INs limits the CCh induced depression of the DG population spike. I realize interpreting these data is difficult in light of the complicated local circuit, however an effort should be made to relate how a loss of NPY-mediated modulation of this depression would result in enhanced background conditioning.

REPLY

Thank you for pointing this out. We have added new experimental data and discuss this issue now more carefully:

1. Neither BIBP3226 nor HIPP silencing affected population spike size in the absence of muscarinergic receptor stimulation. This finding is in line with a recent report (Lee *et al.*, 2016). We show this now in Supplementary Fig. S7b and discuss it on page 17 lines 4-13.
2. Using a Chat-Cre mouse line, we demonstrate that pharmacogenetic stimulation of endogenous ACh release in our slice preparations is excitatory in the DG, and in parallel stimulates NPY-mediated inhibitory transmission through the here identified local circuit. This is evident by the increase in population spike area upon BIBP3226 application, which can be observed only upon activation of cholinergic transmission (Fig. 6).

Based on our circuit dissection, we suggest that HIPP cells mediate an inhibitory ACh response via NPY that is stimulated during background context conditioning. While the selective blockage of HIPP-mediated M1 effects enhances context memory salience, the complete pharmacological blockage of muscarinergic receptors in the DG (including M1 receptors on both HIPP cells and granule cells) blocks it (Supplementary Fig. S9a). We suggest that the outcome of cholinergic modulation of context memory salience depends on the relative weight of direct and indirect (via HIPP) effects on DG granule cells, which may be adjusted by altering excitability of HIPP cells and their NPY expression level (e.g. via CREB). Please see page 18 line 9 – page 19 line 1.

Reviewer #2 (Remarks to the Author):

In this manuscript the authors study the role of hilar interneurons that are associated with the performant path (HIPP interneurons) in the dentate gyrus, in fear conditioning. Using transgenic mice they first show that silencing HIPP interneurons prior to learning using an inhibitory DREADD enhances memory of the background context – that is that associated with the conditioned stimulus, but has no effect on cued memory (tested in a different context). They then go on to show that this effect requires the activity of muscarinic M1 receptors and Y1 NPY receptors. Moreover, it appears that the effect of NPY is mediated by CREB mediated expression of NPY. Overall this is a nice piece of work that establishes the role of HIPP interneurons in the encoding of background context, and the data support their proposal that this encoding requires the activity of the neuropeptide NPY.

The role of the dentate gyrus in pattern separation is well established, and recent experiments using genetic manipulations are showing the key role it plays in context discrimination during fear learning. The involvement of the cholinergic system in this is also well known. Moreover, the role of hilar interneurons in spatial learning is reasonably well established. The new data here are showing the role of HIPP interneurons in context salience, and a suggesting a role for CREB and Y1 receptors.

In this study they show that cholinergic modulation requires Y1 activity there is no indication how these two receptor systems interact, and what the impact of CREB mediated expression of NPY is. It is surprising that a peptide transmitter system is genetically upregulated and it interacts with the cholinergic system but one is left wondering what the impact of this is. Is there more NPY released? Is it only released following fear conditioning and CREB activation? How does NPY affect cholinergic activity.

REPLY

Thank you for pointing this out. To address these questions, we now add (1) an experiment demonstrating that combined inhibition of HIPP cells with hM4Di and NPY receptor blockage does not have a larger effect on oxotremorine M induced inhibition than NPY receptor blockage alone (Supplementary Fig. S7b), confirming that NPY largely mediates the effect of M1 activation of HIPP cells (page 17 lines 4-13). Furthermore, we (2) determined the effect of CREB inhibition on these physiological responses and found a reduced effectiveness of muscarinic inhibition under stimulation with low concentration of oxotremorine M (Supplementary Fig. S8) and discussed this on page 18 lines 9-16.

Other concerns:

1. The impact of cholinergic activation on HIPP cell activity and its overall impact are only shown with bath application of agonists and antagonists. It is entirely unclear what the impact of synaptic stimulation of cholinergic afferents would be. This really needs to be shown as the tools for this are widely available.

REPLY

To address this question, we added an experiment in which in Chat-Cre driver mice we selectively targeted the cholinergic neurons of the medial septum to express the activatory DREADD receptor

hM3Dq. Following transport of hM3Dq protein into synaptic terminals, we were then able to stimulate endogenous ACh release in slice preparations using CNO application to the bath. This stimulation resulted in a moderate increase of DG population spike responses to perforant path stimulation that was further increased through the application of BIBP3226 (Fig. 6; Results page 12 line 24 – page 13 line 12). These data suggest that ACh released in the DG can exert facilitatory effects in line with (Zhang *et al.*, 2010, *J Neurosci*, 30, 6443-6453; Sawada *et al.*, 1994, *Neurosci Res*, 20, 317-322) and that this effect is attenuated by an NPY mediated inhibition. We suggest that HIPP cells as the mediators of ACh-induced NPY transmission in the DG regulate this attenuation effect and limit granule cell activation also during memory formation, as also evident by the increase in the number of cFos⁺ cells upon HIPP cell blockage during background context conditioning (Fig. 1h) (Results page 6 lines 14-16).

2. *When testing the effect on HIPP cells they use oxotremorine, as somewhat selective M1/M2 agonist and show its effect is blocked by pirenzepine. However, for the field potential data they switch to carbachol, which is much less selective, and no antagonist is shown. I suggest that both experiments should be shown using the same agonist and antagonist.*

REPLY

We agree with the reviewer; carbachol was previously chosen with reference to (Kahle & Cotman, 1989, *Brain Res*, 482, 159-163; Burgard & Sarvey, 1990, *Neurosci Lett*, 116, 34-39) and its effects have now been replicated using oxotremorine M application (Fig. 4e).

3. *HIPP interneuron are SST positive, however, there is another population of PV-positive interneurons in the hilus. SST interneurons are known to innervate PV-interneurons. What then is the role of PV-interneurons?*

REPLY

We fully agree that it is important to address the role of different DG interneuron populations in context memory salience and thus performed additional experiment to address this question. Since HIPP cells are known to inhibit PV interneurons (Savanthrapadian *et al.*, 2014), we tried to mimic the effect of HIPP cell inactivation by activating PV⁺ neurons during background context conditioning using hM3Dq receptors expressed in PV-Cre driver mice and observed reduced background context memory (Results page 6 line 19-24, Supplementary Fig. 3). This renders the possibility highly unlikely that HIPP cells increase GC activation through a disinhibitory mechanism via PV cells. However PV cells may be recruited during the regulation of background context memory independently of HIPP cells, and HIPP cell mediated effects on the firing precision of PV cells may still be involved. Please see page 14 lines 1-6.

Reviewer #3 (Remarks to the Author):

Overall

In this manuscript, the authors demonstrate that hilar performant path-associated (HIPP) cells of the DG mediate the devaluation of background context memory during fear conditioning. The authors reduced NPY expression and blocked NPY-Y1 receptors in the DG to show this sensitivity. Moreover, they show that M1 muscarinic receptors mediate the cholinergic activation of HIPP cells and mediate HIPP cells' control of background context salience. I have concerns with the figures and statistical analysis. Some additional control experiments are needed.

Major Comments

Results

1. The authors should include all data, especially when conditioning was done without any CS presentation (page 5, line 104). The stats should not just be recorded. The data should be included at least in supplement. See also page 5, line 109; page 8, line 168; etc.

REPLY

We have included the data as advised by the Reviewer in Supplementary Fig. S2a.

2. The authors should elaborate or hypothesize why both viral vectors increased SST expression (page 6, line 133).

REPLY

Only the expression of CREB^{S133A}, but not native CREB, results in a significantly increased SST expression, this has now been corrected in the figure legend to Fig 2d. Although S133A mutants have been successfully used to suppress CREB function in learning experiments, it should be considered that CREB activation could also occur without S133 phosphorylation. Recently, Asahara *et al.*, 2001 (Molecular and cellular biology, 21, 7892-7900) showed that stimulation of SST promoter on naked DNA enhances expression independently of its phosphorylation at S133. However, recruitment of p300 depends on S133 phosphorylation and p300 stimulates SST transcription on chromatin in an S133 phosphorylation dependent manner. Thus, our findings of a mild increase of SST after overexpression of CREB^{S133A} and tendentially similar after native CREB are in line with a constitutive regulation mode and potentially decompacted chromatin at the SST gene in the HIPP cells of our mice. By contrast, NPY and GAD appear to be regulated in an inducible manner via S133 phosphorylation. This supports the notion that CREB action is gene specific and likely depending on chromatin compaction and plays multiple roles in the adjustment of HIPP cell function. However, the change of SST expression is small compared to the regulation of NPY. Please see page 7 lines 16-25 and page 16 lines 15-19.

3. The authors administer a BIBP3226 injection before retrieval (page 8, line 168). This type of experiment should be performed in all figures. To assess whether these cells are necessary for encoding and/or retrieval is essential. For example, in figure 1, CNO was given before acquisition.

Why isn't the CNO additionally given in a separate cohort of mice before retrieval? These experiments should be included.

REPLY

Thank you for this remark. We performed the requested experiment and silenced HIPP cells before background memory retrieval (Supplementary Fig S2d; Results page 6 lines 4-6). The silencing of HIPP cells before contextual and cued retrieval was without any effect as compared to the control group, in line with the results of our BIBP3226 experiments.

*4. As an example, the authors state that *Chrml* is not observed in GFP- hilar neurons or GFP+ hippocampal neurons. The authors give the ANOVA, but not present the follow up *p* values for the *t*-tests or Fisher's LSD. This is consistently a problem throughout the paper. Moreover, in the aforementioned figure, if the GFP+ neurons are indeed significantly different from the other two groups, why is there only one asterisk in that panel in Figure 4A? Why is the hilar GFP+ v. hippocampal GFP+ difference not presented? I highly suggest that the authors create a table with all statistical analyses performed, *p* values, follow up tests, etc. As the manuscript is right now, the statistical analyses need major improvement. (please see page 9, line 192; Figure 4a, panel 2).*

REPLY

We have corrected the *P* values in Fig. 4a and Supplementary Fig. S6b. To meet the referee's criticism, we are now providing a complete table of all statistical analyses as Supplementary Table S1. In addition, all ANOVA values are presented in the main text and pairwise comparisons are shown in our figures.

5. The authors should also consider order effects. I recommend performing at least Figure 1b and reversing the order of shock context and CS presentations (e.g. CS presentation on day 15 and shock context on day 16). It would further increase the validity of the increased fear expression in the background group.

REPLY

We completely agree that order effects must be considered. In fact, with our focus on the contextual memory salience we tried to avoid any potential confound through prior cued retrieval and therefore always performed contextual retrieval tests first. To directly address the referee's concern, we now tested C57BL/6 mice with reversed order of retrievals, but could not observe evidence of an order effect in our paradigm. We include this information here as a Referee Figure 2.

Referee Figure 2. C57BL/6 mice trained with background context conditioning. In retrieval session, mice that are tested first for the cued memory ($n=8$ CS first group) (as our normal paradigm) show no difference to the group of mice that are tested for shock context (similar to our paradigm; $n=8$ shock context first group). Data are mean + s.e.m.

6. *Figure S4: The authors should more fully discuss the increased post-training freezing in the experimental group. This is problematic. Moreover, I recommend scoring the shock reactivity (distance traveled during the shock). Do the experimental mice respond differently to the shock?*

REPLY

Using icv injection of BIBP3226, we could rule out a potential bias through anxiogenic effects of this drug. Moreover, we have now analysed the distance traveled during the shock presentation as per request of the Reviewer, indicating no difference in shock sensitivity and responsiveness. Thus, the observed effect may most likely relate to enhanced immediate context memory (Fanselow, M.S, 1986, *Learning and Motivation*, 17, 16-39). The effect is also seen with M1 receptor knock down, but not with general inhibition of HIPP cells via hM4Di receptors or dominant negative CREB. We now discussed in the legend to Supplementary Fig. S5a and Supplementary Fig. S9d, and on page 16 line 22 – page 17 line 3.

We include the data of activity during shock here as a Refree Figure 3.

Referee Figure 3. Distance travelled during the presentation of CS-US pairings in reference to Supplementary Fig S5a (a) and Supplementary Fig. S9d (b). Distance travelled by mice during the presentation of sound, delivery of E-stimulus and inter stimulus intervals is unchanged between the groups in both the figures. Data are means +sem.

7. Figure S6: Why is the 2.5 $\mu\text{g}/\mu\text{l}$ scopolamine omitted from the EPM analysis? This group should be included.

REPLY

We are sorry for this error. We have now included the 2.5 $\mu\text{g}/\mu\text{l}$ scopolamine group, which produced a mild anxiolytic-like effect. This is explained in the Results on page 12 lines 5-9 and in the legend to Supplementary Fig. S9b.

8. Figure S6: Moreover, the authors write: “both doses further reduce background fear memory but not auditory cued fear memory compared to vehicle-injected controls.” In panel A, the 2.5 $\mu\text{g}/\mu\text{l}$ scopolamine is significantly less than vehicle during the CS test, but the 5.0 $\mu\text{g}/\mu\text{l}$ scopolamine is not. Something is incorrect in either the figure legend or in the stats presented in panel A.

REPLY

Thank you for pointing this out, the figure legend was incorrect since ANOVA revealed no significant effect of scopolamine on auditory cued fear memory. This has been corrected now in Results on page 12 lines 2-5 and legend to Supplementary Fig. S9a.

9. The authors should clearly explain the rationale of the experiments on page 9: the passage from oxotremorine to carbachole is not clear, they are both AchR agonist but they act on different

receptors, also in the first case they investigate the spike frequency in the second case the population spike area. Moreover, they should indicate the exact values for Fisher's LSD one-way ANOVA, Wilcoxon signed rank test, and paired t-test.

REPLY

We have addressed this point by reproducing the effect of HIPP cell blockage and BIBP3226 upon stimulation of muscarinergic receptors with oxotremorine M (Fig. 4e). Further we investigated spike frequency to directly demonstrate the regulation of HIPP cells via M1 receptors, and population spike area size to determine their effect on DG excitation. Please see page 31 lines 19-20 and page 29 lines 14-15. We provide the detailed statistics in Supplementary Table S1.

Minor Comments

Introduction

1. The authors discuss the sufficiency of engrams in the DG for contextual memory, but completely omit the necessity work on DG engrams. The authors should include relevant literature on both necessity and sufficiency in the engram paragraph as well as in other relevant areas.

REPLY

Thank you very much for this remark. A discussion of the necessary hippocampal subregions is now included in the introduction on page 4 lines 5-11.

2. The authors state that the size of the activated DG granule cell ensembles correlates with contextual memory strength, but again omit the work performed on CA3. Their overview of engrams is overly simplistic and should consider the output of the DG to CA3 in their strength of a given memory discussion.

REPLY

We have now included references concerning the involvement of CA3 in context memory encoding in the introduction on page 4 line 5-11 and discussed the cholinergic modulation of its function in this context on page 17 line 23 – page 18 line 8.

Methods

3. Page 16, line 366: there is a word missing before allowed. The sentence does not make sense as it.

REPLY

The sentence has been corrected.

4. Why are the mice single caged before the start of experiments? This is social isolation and is not appropriate for the experiments. The authors should perform all experiments in group housed mice

for future experiments. I am not suggesting that all experiments be redone for this manuscript, just that they change this protocol for future experiments.

REPLY

Thank you very much for this comment; we will follow your advice in our future experiments. We would like to point out that we tried to avoid extensive isolation of animals and kept them separately for only few days before and during the behavioral experiments.

5. Why do the authors administer 2, 4, and 8 mg ip. These doses seem incredibly high. Why not administer 2-3 mg per day for 5 days to induce recombination? Was this previously published?

REPLY

We have systematically established the protocol for tamoxifen induction in our lab in order to obtain strong and lasting recombination without inducing adverse effects on the animals general condition and found this protocol more efficient to induce recombination in SST⁺ neurons than the protocol reported originally for this mouse line (Taniguchi *et al.*, 2011, Neuron. 71, 995-1013). Robust and lasting recombination has been described in other mouse lines with high concentration induction protocols (Madisen *et al.*, 2010, Nat Neurosci, 13, 133-140: 6 mg/day for 5 days; Reinert *et al.*, 2012, PloS One, 7, e33529: 3x8 mg spaced over 5 days), but we refrained from using them in our behavioral experiments as we observed signs of nausea and reduced wellbeing in our animals with such high doses. This is now pointed out in Methods on page 25 lines 19-22.

Figures

6. Why is each panel not referenced in the text? For example, Figure 1D-1E are not referenced. Each panel should be referenced.

REPLY

We make sure now that every panel is referenced in the text. Fig. 1d and 1e are now referenced on page 5 line 18 and 21.

7. Figure 2C: The authors state that the two experimental groups are different than the control group for SST. However, only 1 comparison line is shown. The authors should report both post-hoc tests and show two separate lines from control to experimental group as they did in the NPY group.

REPLY

Although both treatment groups induced similar expression levels, the difference to control only became significant for the CREB^{S133A}. The statement in the figure legend has been corrected.

8. *Figure 2: An experimental schematic should be included.*

REPLY

We have included a schematic as suggested.

9. *Figure 3: An experimental schematic should be included. Furthermore, Figure S4 and Figure 3 should be combined into 1 main figure.*

REPLY

We have transferred the schematic from previous Fig S4a to Fig. 3 and included the memory data from previous Supplementary Fig. S4c. We left training related data in the supplemental information, consistent with our other experiments.

10. *Figure 5C and 5D: Again, these panels are not referenced.*

REPLY

Figure 5c and 5d are now referenced in the results section on page 12 line 14 and 16.

11. *Figure S1A: I suggest moving this schematic to Figure 1. It would be useful for readers.*

REPLY

We have included previous Supplementary Fig. S1 to Fig. 1 now.

12. *Fig. 4C: The authors should add a graph showing the spike frequency (In the first big representative trace, the increase in the spike number is not so evident like in the single representative traces below). I suggest they can put together Fig 4C and S5D (panel 3).*

REPLY

Thank you for these suggestions, we changed the figure as advised.

13. *Fig. S5D: The authors should specify the duration and intensity of the current pulses they used to elicit spike frequency.*

REPLY

We now provide the details of current injection in the Methods under section patch clamp recordings page 32 lines 23-25.

14. Figure S5D: In the legend to panel D, there is a spelling error for pirenzepine.

REPLY

Thank you very much. The error has been corrected.

Reviewers' Comments:

Reviewer #1 (Remarks to the Author):

I find the manuscript of Raza et al to be greatly improved with the additional experiments, analysis and discussion. I am satisfied with the authors' response to my comments, as well as the comments of the other reviewers, and support its publication.

I only have one minor issue to point out: On the bottom of page 5 (line 113/114) they write "Thus, HIPP cell activation abolished the differences....", I believe it should read "Thus, HIPP cell inactivation..."

Reviewer #2 (Remarks to the Author):

My apologies for taking so long over this. In the revised manuscript, the authors have addressed all the issues that I raised by doing more experiments and modifying the text and discussion. Overall, its a much improved and strong paper that presents a very complex model on how pattern separation in the dentate gyrus during contextual fear conditioning separate background from foreground. This network engages several types of neurons in the dentate dentate gyrus, cholinergic input and trascriptional changes in NPY signalling. Having read the paper several times, I have to say I still do not have a full understanding of how these elements lead to pattern separation of these two issues. I fear that this will also be the case for the reader of this paper and I would encourage the authors to include a schematic in the discussion describing the different parts of this circuit and their engagement during learning and recall.